# Physical Measures of Welfare in Fin (*Balaenoptera physalus*) and Humpback Whales (*Megaptera novangliae*) Found in an Anthropized Environment: Validation of a First Animal-Based Indicator in Mysticetes

**DOI:** 10.3390/ani14233519

**Published:** 2024-12-05

**Authors:** Anik Boileau, Jonathan Blais, Marie-Françoise Van Bressem, Kathleen E. Hunt, Jamie Ahloy-Dallaire

**Affiliations:** 1Faculté des Sciences Animales, Université Laval, Québec, QC G1V 0A6, Canada; jamie.ahloy-dallaire.1@ulaval.ca; 2Centre d’Éducation et de Recherche de Sept-Îles, Sept-Îles, QC G4R 2Y8, Canada; jonathan.blais@cegepsi.ca; 3Cetacean Conservation Medicine Group, Peruvian Centre for Cetacean Research, Museo de Delfines, Lima 20, Peru; mfb.cmed@gmail.com; 4ProDelphinus, Miraflores, Lima 18, Peru; 5Department of Biology, Smithsonian-Mason School of Conservation, George Mason University, 1500 Remount Rd, Front Royal, VA 22630, USA

**Keywords:** humpback whales, fin whales, welfare assessment, anthropized environment, chronic stress, entanglement, physical states, body condition, cutaneous lesions

## Abstract

The welfare of free-ranging cetaceans is being impacted around the world due to human activities like commercial fishing and marine traffic. Here, we validate a non-invasive physical indicator of welfare for humpback and fin whales found in the Gulf of St-Lawrence, Canada. Multi-scale measurement of welfare showed that most humpback whales were in a good welfare state, whereas most fin whales were in a moderate state. For both species, welfare was mainly affected by prolonged effects of physical trauma (entanglement in fishing gear and boat collisions) on body condition, specifically whales being thinner. Other factors impacting welfare included degradation of the marine environment (e.g., oxygen levels, water temperatures and salinity levels).

## 1. Introduction

In recent years, anthropogenic disturbance has had an increasing impact on the welfare of wild cetaceans around the world [1,2,3,4,5,6,7]. Marine debris, entanglement in fishing gear, ship collisions, underwater noise, whale-watching activities and degradation of the marine environment are currently the main welfare issues identified by international conservation bodies and marine mammal research groups [6,8,9,10,11,12]. The welfare of an animal can be defined as “*its state as regards its attempt to cope with its environment […] it refers to the extent of biological activity underlying attempts to cope, including the involvement of body repair systems, immunological defences and physiological stress responses as well as a variety of behavioural responses*” [13]. Furthermore, the “welfare state” refers to the underlying emotional status of an animal, characterized by positive or negative valence, arousal, duration and generalization of emotions, where it “*constitutes an internal, central (as in central nervous system) state*” [14]. This recent interest in the welfare of individual animals underlines new ethical concerns about cetaceans, recognizing them as sentient beings that can experience subjective affective states such as pain and stress due to human activities [15,16,17,18]. Interest in free-ranging cetacean welfare also represents a complementary approach to classical conservation biology [19,20,21], which, together, constitute a continuum of ethical concerns that range from the individual’s welfare to the conservation of a healthy population. In this regard, legislators have been urged to include cetacean welfare assessment protocols in regulation and conservation plans, notably by the Whale Killing Methods and Welfare Issues working group of the International Whaling Commission [8,9,10,22,23,24].

Located in the Atlantic Ocean, the Gulf of St-Lawrence (“St-Lawrence” hereafter), Canada, is a highly anthropized environment and is a well-known feeding ground for rorquals like the humpback (*Megaptera novangliae*) and fin whale (*Balaenoptera physalus*) [25,26,27,28]. Each year, in early spring, these two species enter the St-Lawrence to feed on zooplankton and small pelagic fishes [29,30,31] for several months before migrating out of the Gulf, usually before ice cover occurs [32,33]. However, due to extensive human activities, predominantly commercial fishing and maritime traffic, the animals feeding in the St-Lawrence face important welfare challenges that could have dire consequences at the individual and population levels. To truly understand the human impact on these whales, methods to assess welfare are needed. In this regard, our epistemological posture is based on multiple theoretical frameworks developed for structuring coherent assessment protocols (e.g., the Five Freedoms Model [34,35] and the Five Domains Model [36,37,38]), which have been applied to farm [39,40], companion [41,42], captive wildlife [43,44], laboratory animals and, more recently, to free-ranging species impacted by anthropogenic activities [45,46,47,48,49]. The common welfare assessment principles we adhere to include the necessity to engage in a holistic approach to identify relationships between animal-based indicators of welfare (physical, physiological and behavioural indicators, which include their respective measures) and environmental parameters that positively or negatively impact the affective state of each animal [22,36,38,50,51,52,53].

The first indicator to be validated in a welfare protocol is usually the physical status of an animal. Depending on the species and the context, the protocol may include a wide range of measures such as body condition score [54,55,56], lameness score [57,58], number and severity of injuries [59,60] and clinical observation of diseases [34,61]. Once the selected measures comprised in a physical indicator are validated statistically, they can be used to infer the underlying affective state through a scoring system [38,52,62]. On its own, the physical indicator can partially inform on the overall welfare of an animal but needs to be integrated into a complete species-specific welfare assessment protocol (e.g., the Five Domains model by Mellor [38] or the Welfare Quality^®^ protocols [39]).

In captive cetaceans, one welfare assessment protocol was developed for bottlenose dolphins (*Tursiops truncatus*) based on four welfare principles, 11 welfare criteria and 36 different measures including 9 specifically assessing physical state, e.g., body weight, injury and diseases [63]. A scoring system based on a 0–2 scale (0 = good welfare, 1 = adequate and 2 = poor welfare) was used to assess all measures. In the context of free-ranging cetaceans, the assessment of animal-based measures is particularly challenging due to remoteness and other logistical difficulties of research at sea (e.g., weather and sea state limitations), which result in limited time of direct observation. However, conservation medicine and physiology approaches have recently validated four visual measures of health that could be used in a physical indicator of welfare for free-range cetaceans: body condition, prevalence of injuries, cutaneous lesions and parasite and/or epibiont loads [7,64,65,66,67,68].

Body condition has been extensively validated as an informative measure of welfare in mammals, since chronic states of energy deficit lead to endocrine metabolic homeostasis dysregulation, clinically observed by weight gain or loss [69,70,71]. For instance, emaciation can reflect poor nutritional status, chronic stress and underlying pathologies [72,73,74,75], which can be emotionally experienced as feelings of hunger, pain and discomfort [38,50,76,77]. However, body weight fluctuations can also reflect reproductive status, age and seasonal migration for some species, which should be considered when developing species-specific scoring systems [78,79,80]. In North Atlantic right whales (*Eubalaena glacialis,* “NARW”), a three-point scale for body condition was developed in 2004 and remains the reference for baleen whale visual body condition scoring system. This scale has been useful for visually assessing health in NARW for over three decades, ultimately showing a decline in overall physical health over this time period [81,82,83,84]; further, poor scores are known to be predictive of subsequent reproduction and mortality. Similar approaches to assessing body condition as a visual measure of health have since been validated for southern right whales (*Eubalaena australis*) [85], grey whales (*Eschrichtius robustus*) [86], blue whales (*Balaenoptera musculus*) [80], common dolphins (*Delphinus delphis*) [87], and short-finned pilot whales (*Globicephala macrohynchus*) [88]. However, body condition evaluation has only been partially examined in humpbacks [7], and has not been applied to fin whales.

Besides body condition, physical injuries of free-ranging cetaceans have also been measured visually by characterizing open wounds, deformities and scars using either unmanned vehicles, boat-based photographs or through post-mortem investigation [7,89,90,91,92,93]. Some scoring systems were developed based on the location of injury on the body, type and cause of injury (e.g., entanglement, boat strike, etc.) and severity of the observed lesions [90,94,95]. For instance, analysis of boat-based photographs collected in the St-Lawrence (2009–2016) showed that 6.5% of fin whales (n = 322) and 85% of humpback whales (n = 112) had visible scars from previous entanglement [67]. However, fin whales rarely lift their flukes fully out of the water (as humpback whales do), thus limiting the body areas available for assessment and limiting accurate assessment of entanglement rates [67]. From a welfare standpoint, physical injuries, past and present, have a detrimental impact on the animal by inflicting pain and suffering but also, depending on the location and severity, can negatively affect survival and fitness (e.g., half a fluke missing). The level of stress resulting from physical pain and injuries can have prolonged effects on health and welfare [96,97,98].

Cutaneous disease may also provide indications about the health status of cetaceans. For instance, prolonged freshwater exposure in humpback whales and oceanic dolphins may lead to ulcerative dermatitis and, ultimately, death [99,100]. As another example, in bottlenose dolphins (*Tursiops* spp.), the high prevalence of lobomycosis, a fungal granulomatous skin disease is indicative of environmental degradation [68,101,102], and the epidemiological pattern of tattoo skin lesions caused by poxviruses may be indicative of poor health and stress [102,103]. Although complete diagnosis can only be achieved through histology and immunohistochemistry of skin samples, visual assessments based on previously diagnosed skin pathologies have been validated for multiple types of cutaneous lesions [100,104,105]. Most of the scoring systems used to quantify cutaneous lesions indicate presence or absence, in addition to percentage of the body covered and lesion size [81,102,106,107].

The presence of ectoparasites and epibionts on the bodies of whales and dolphins can also reflect health status since diseased or injured individuals will often reduce their swimming speed, favouring epibiotic settlement [108,109]. The relationship between the epibiotic organisms and their cetacean hosts can be one of parasitism or commensalism and can have different consequences on the overall welfare of the whales [108,110,111]. When covering extensive parts of their hosts, diatom algae and barnacles like *Coronula diadema* are thought to impact swimming drag and hydrodynamics [109,112,113,114], but have a relatively small impact on overall welfare [110]. In comparison, cookiecutter shark bites (*Insitus* sp.), characterized by considerable skin removal, will leave open “cookiecutter shape” wounds that can range between 8 and 18 cm in diameter [115,116,117]. The welfare impact of such open wounds does not only concern physical pain, but also susceptibility to infectious diseases and other sequelae [118,119,120].

Recent studies on the epidermal condition of fin and humpback whales have reported numerous types of lesions originating from infectious diseases and ectoparasites. Fin whales found in Antarctic feeding grounds showed a high prevalence of cookiecutter shark bites compared with fin whales observed in the Strait of Gibraltar, where the highest prevalence of cutaneous lesions was associated with the copepod ectoparasite *Pennella balaenopterae* [73,121]. In the endangered humpback whale population found in the Arabian Sea, 41% of animals (n = 93) exhibited tattoo skin disease-like lesions (TSD-L), possibly caused by cetacean poxvirus [7], compared with humpbacks found in Icelandic waters, where the prevalence of TSD-L was only 0.7% (n = 728) [122].

When comparing visual health assessment results across species of cetaceans found in different geographical areas, we can begin to evaluate the effects of anthropogenic activities and marine environmental degradation on cetaceans. Therefore, the development of a standardized method to assess the physical state and welfare of more species of free-ranging cetaceans is needed. The aim of the current study was to validate a physical indicator of welfare for humpback and fin whales observed in a seriously anthropized feeding ground. We hypothesized that the four different animal-based measures—body condition, injury condition, skin condition and parasite/epibionts condition—would show convergent validity, supported by these measures loading onto a shared factor in common factor analysis. Additionally, we theorized that these measures would show discriminant validity, supported by only low to moderate positive correlations between them. Furthermore, we assumed that normal environmental indices (lower water temperature, higher salinity levels, higher oxygen and lower nitrate levels) would correlate with good skin health and lower parasite/epibionts loads, and prey availability would correlate positively with body condition scores. The findings of this study could contribute to the development of wild cetacean welfare assessment protocols, which could be valuable in informing policymakers and government agencies on conservation priorities for these species.

## 2. Materials and Methods

The present study is part of a larger ongoing project on the development of an integrated welfare assessment protocol for humpback and fin whales found in the Gulf of St-Lawrence. The project is in compliance with the Canadian Council on Animal Care guidelines [123].

### 2.1. Fieldwork Methods

Aboard a research platform (19-foot inflatable rigid hull boat with a 90 hp outboard motor), a focal study approach was used to collect photographic data on humpback and fin whales observed in the Sept-Îles area from 2016 to 2021 (Figure 1). Once an animal was observed, we approached to position the boat parallel to the whale’s body and took photographs of the animal (all possible angles) at ≥50 m distance for photo identification purposes and physical state characterization (Nikon D-800; 500/1000 mm and 18/200 mm lens). The maximum time spent with an individual was two hours, including diving and breathing sequences. If an animal actively avoided the boat (e.g., dives immediately when the boat approaches), the focal study was terminated.

### 2.2. Image Treatment and Analysis

All images were treated with Photoshop (2021) to standardize and maximize quality in accordance with validated photo-identification methods [124,125]. After analysis and categorization (e.g., matching the individual whale with our local catalogue), photographs of the fluke of each humpback whale and the right-side chevron and dorsal fin of fin whales were uploaded into an online database (Happy Whale) to increase photo-identification capture–recapture information and collaborative efforts [126]. For repeatability and validation purposes [80,81,86], a single observer (the first author, a researcher experienced in field whale research and welfare assessment methodologies for multiple species), analyzed the image dataset of the 50 first individual humpbacks and 50 first individual fin whales encountered between 2016 and 2021 for which at least 10 high-quality photographs (at least one of each side of observable body parts in both species and fluke for humpbacks) were taken during the focal observation. If a whale was resighted on multiple days/years, only the photographs of the first encounter were used for the welfare assessment.

### 2.3. Environmental Measures

Environmental parameters were retrieved from Fisheries and Oceans Canada’s open database (https://search.open.canada.ca/data/?sort=metadata_modified+desc&search_text=Salinity+levels+St-Lawrence&page=1 (accessed on 4 December 2024)) and included sea surface temperatures (°C) and salinity levels (%) [127], bottom temperatures (°C) and salinity levels (%) [128], bottom dissolved oxygen content (% saturation) [129], mean integrated chlorophyll-a (mg/m^2^) measured at depths between 0 and 100 m [130], integrated nitrate (mmol/m^2^), phosphate (mmol/m^2^), and silicate (mmol/m^2^), measured between 0 and 50 m and from 50 to 150 m [129]. These abiotic measures were collected from different sites throughout the St-Lawrence, notably from the Rimouski station, located upstream and the Shediac Valley area in New Brunswick, downstream just outside the Gulf of St-Lawrence. The data were collected once a month at each site. We averaged the datasets across these sites since whales observed in the Sept-Îles area are mobile throughout the entire St-Lawrence and western parts of the Atlantic Ocean. These environmental parameters can inform the level of eutrophication (impact of urban runoff, agricultural fertilizers, human waste, etc.). We also retrieved annual biotic measures (same website, collected in the Rimouski area, Anticosti, located in the Gulf of St-Lawrence, and Gaspé, on the south shore of the Gulf) on total zooplankton wet weight (g/m^3^) [129] and pelagic fish species abundance for sand lances (*Ammodytes* sp.), Atlantic herring (*Clupea harengus*), capelan (*Mallotus villosus*) and Atlantic mackerel (*Scomber scombrus*), which are the main dietary items of humpback and fin whales [128,131,132,133].

### 2.4. Physical Measures and Welfare Scoring System

Based on existing validated welfare assessment protocols in other species and in an intra-individual approach [7,39,46,63,134,135], we identified four categories of physical measures that could be assessed for both humpback and fin whales: body condition, skin condition, injury/scarring condition and parasite/epibiont condition. A scoring system was developed consisting of four different scales, one for each physical measure (three scales 0–4, and one scale 1–3) (Table 1). We used the Criteria Importance Through Inter-Criteria Correlation method (CRITIC) to determine the weight of each measure [136,137]. This method is used to combine multiple criteria into one overall score by attributing a larger weight to the measures that have a greater standard deviation, and that are negatively correlated to other pairs of criteria. The method comprises five steps: normalization of the criterion matrix, calculation of the standard deviation of each criterion, creation of the correlation matrix using a pairwise comparison between the criteria, calculation of H index and finally, calculation of the weight for each criterion (see Krishnan et al. [138]). We used the CRITIC method separately for the fin and humpback whale measurement scales, as the results showed different weights should be used for each criterion in each species (Table 1). The final percentage grading scale was used to infer (arbitrarily based on the worst and best scores) the overall welfare state of each animal, as follows: a score between 1 and 40% was considered to represent poor welfare, a score > 40% up to 75% was considered moderate welfare, and a score > 75% was considered good welfare (see Table 1).

#### 2.4.1. Body Condition Scoring

The body condition scoring scale was subdivided into three categories used to generate one of five possible scores. First, whales were categorized as either emaciated, thin/moderate or optimal/good condition. These categories were developed based on protocols validated in other species [80,81,86]. The second step was to attribute a score to the animals, which took the date of observation into account. As whales usually arrive in the St-Lawrence to feed in May or June, body fat stores typically increase by mid-July. Emaciated animals were given a score of 0–1 (where 0 is attributed to whales observed after mid-July and 1 before mid-July); thin/moderate, a score of 2 or 3 (again, 2 after mid-July and 3 before mid-July) and a score of 4 was attributed to optimal weight condition independent of the date (Figure 2). Body condition assessment of females accompanied by a calf is always either emaciated or thin due to high energy requirements during lactation; thus, an increase of 1 point was attributed. For example, a lactating female with a body score of 0 was increased to 1 to compensate.

#### 2.4.2. Skin Condition Scoring

The skin condition scoring scale was developed during image analysis and characterization based on existing cutaneous lesion categories in multiple cetacean species [106,107,139] (See Table 2 and Figure 3 and Figure 4). Skin condition was assessed with a scoring scale ranging from 0 to 4: 0 = ≥2 skin lesion types covering ≥ 75% of the observable body surface; 1 = 1 skin lesion type covering ≥75% of the observed body surface; 2 = ≥1 skin lesion type covering ≥ 25% and ≤74% of observed body surface; 3 = ≥1 skin lesion type covering ≤24% of the observed body surface and 4 = ≤5% of observed body covered in skin lesions (Figure 3).

#### 2.4.3. Injury and Scar Condition Scoring

Based on previous studies, we developed a scoring system for physical injuries and scarring with a 0–4 scale [67,91,141,142,143,144,145,146]. A score of 0 was attributed to whales that had an important physical injury of anthropogenic origin (typical entanglement or boat collision wounds or scars; see Robbins and Mattila 2001 for details [144]) that could alter swimming, diving and/or thermoregulation, like fluke and dorsal fin amputation (Figure 5A). A score of 1 was attributed to animals that had open vascularized wounds (Figure 5B) and a score of 2 for deep healed injuries (no vascularization observed but characterized by a visible dent in the epidermis, Figure 5C–E), both with anthropogenic origin characteristics. A score of 3 was given for superficial healed anthropogenic injuries (scars with no dent in the epidermis Figure 5F,G) or important natural scarring from either killer whale predation attempts (visible tooth rake scars, Figure 5H) or agonistic interactions with conspecifics (multiple scars from different angles, typical in male humpback whales). Finally, a score of 4 was given to animals that did not have any visible scars from anthropogenic origins but could have multiple superficial scars from natural origins.

#### 2.4.4. Parasite and Epibiont Condition Scoring

The parasite and epibiont condition scores were included in the same category, although their presence or the lesions they cause might not affect the whales equally (Table 3). The scoring system was developed using a 1 to 3 scale, where 1 = ≥40% of the observable body was covered by parasites/scars and/or epibionts, 2 = between ≥5% and ≤40%, and a score of 3 = ˂5% (Figure 6 and Figure 7). Barnacle load, only found on the flukes of humpbacks, was assessed by comparing and adding left and right lobe loads and was included in the overall score.

**Table 3 animals-14-03519-t003:** Description of parasite and epibiont categories used for scoring.

Parasite and Epibiont Categories	Description
Orange film/diatoms (OFD)	Orange film covering parts of the body to different extent (Figure 7A) [147,148].
Lamprey bite and skid marks (LBM)	Circular marks with a pale centre and darker edges, sometimes raised. Skidding marks are often observed with parallel scratches (Figure 7B) [139,149].
Cookiecutter wounds/scars (CCW)	Oval-shaped crater-like wounds. Colour will depend on the healing stage, from pink to red in a fresh wound to whitish in a healing phase, to a depressed scar with normal pigmentation once healed (Figure 7C) [149,150].
Whale lice (Louse) (WL)	Small crustaceans of the Cyamid family found on body parts with folds (e.g., the ventral grooves or around blowholes). Infestations can also be observed on open wounds or immune-compromised individuals (Figure 7D) [108,151].
Fluke barnacle load (FBL)	Crustaceans from the family Coronulidae typically attached to appendages like pectoral fins or fluke (Figure 7E,F) [113,152].

#### 2.4.5. Statistical Analysis

Data were analyzed with JMP^®^, 17 (SAS Institute Inc., Cary, NC, USA, 1989–2024) and R 4.1.0 (R Core Team, Vienna, Austria, 2021) with each whale considered as a statistical unit (n = 50 humpbacks and n = 50 fin whales). A total of 6403 images were analyzed (n = 3251 for humpbacks and n = 3152 for fin whales). The scoring systems (the four measurement scales) were each tested with Cronbach’s alpha standardized test for internal consistency, where alpha (α) needs to be greater than 0.7 for reliability of the measurement scales [153]. Discriminant validity between the four measurement scales (body, skin, injury and parasite/epibiont) of our physical welfare assessment protocol was tested with Spearman correlations within each species. Discriminant validity is supported if correlations between two scales are less than 0.75 [154]. We performed a common factor analysis (Q-type) using Maximum-Likelihood and Varimax orthogonal rotation to test convergent validity between the measurement scales of humpbacks and fin whales (standardized loading values greater than 0.4 are considered significant) [155,156]. We also used Spearman correlations to test the relationships between all measurement scales and environmental parameters (temperature, salinity, dissolved oxygen, nitrate, phosphate, silicate, chlorophyll A and biomass of fishes and zooplankton) and dates (day-month-year). We also used the results to identify predictors and outcome variables within each common factor (Figure 8); injury condition predictor on body condition outcome, and parasite/epibiont condition predictor on skin condition outcomes. These were tested with ordinal logistic regressions. Point-biserial correlations were used to test the relationships between nominal variables (presence or absence of cutaneous lesion types, parasite/epibiont species, and anthropogenic injuries) and environmental parameters (temperature, salinity, dissolved oxygen, nitrate, phosphate, silicate, chlorophyll A and biomass of fishes and zooplankton). Furthermore, an ordinal logistic regression model was used to test the likelihood of environmental parameters (fixed effects) influencing cutaneous lesion prevalence (skin disease condition, parasite/epibionts condition) and body condition. These fixed environmental effects were previously transformed to obtain the mean values for the month each animal was observed and the two months prior to observation to account for long-term environmental effects.

Once the CRITIC method was applied to the different measuring scales (see Section 2.4 for the method), a Generalized Linear Mixed Model was used to test the environmental parameters as fixed effects (temperature, salinity, dissolved oxygen, nitrate, phosphate, silicate, chlorophyll A and biomass of fishes and zooplankton) on overall welfare state (random effect was attributed to date (day-month-year).

## 3. Results

Cronbach’s alpha standardized test was significant for both fin (α = 0.76) and humpback whales (α = 0.82), confirming reliability. The scoring scales developed for assessing body condition, skin condition, injury condition and parasite/epibiont condition measures correlated (positively) with each other at low to moderate levels in humpback whales: r = 0.23–0.54 (average r = 0.38) and fin whales: r = 0.10–0.51 (average r = 0.29), showing discriminant validity between the four measures (Figure 9). Common factor analysis showed significant (>0.40) standardized loading factors for each scale in both fin whales and humpback whales (Figure 8). For both species, body condition and injury condition loaded onto a common factor, suggesting convergent validity between these two scales. Similarly, skin condition and parasite/epibiont condition loaded onto a common factor in both species.

### 3.1. Anthropogenic Injuries and Body Condition

The mean body condition score was found to be moderate for both fin (*M* = 69.50, *SD* = 23.50) and humpback whales (*M* = 72.50, *SD* = 19.72). A total of 25 individual (50%) fin whales and 24 humpback whales (48%) had scars originating from anthropogenic activities, specifically fishing gear (n = 46) and small vessel strikes (n = 3). A moderate, yet significant relationship was observed between body condition and injury condition scores in fins (*r* = 0.51, *p* ˂ 0.001) and humpback whales (*r* = 0.54, *p* ˂ 0.001). Moreover, ordinal regression results showed that injury condition scores, originating from anthropogenic activities, significantly predicted poorer body condition in both fin whales (ß = −2.19, SE = 0.64, Wald x^2^ (4) = 14.03, *p* = 0.0006) and humpback whales (ß = −2.63, SE = 0.72, Wald x^2^ (3) = 15.24, *p* = 0.003) (Figure 10). Physical injuries were more severe in 2017 compared to 2020 for humpbacks (*r* = 0.86, *p* ˂ 0.001) and fin whales (*r* = 0.32, *p* = 0.02). Finally, 6% of humpback whales had scars due to orca whale predation (*Orcinus orca*), and none was observed in fin whales.

### 3.2. Parasite/Epibiont Loads and Skin Health Condition

A moderate relationship between parasite/epibiont condition and skin health condition was observed in fin whales (*r* = 0.45, *p* = 0.002) and humpback whales (*r* = 0.53, *p* ˂ 0.001), and ordinal regression showed that animals with parasites and epibionts were more likely to have poorer skin health in both species (fin whales ß = −2.22, SE = 0.85, Wald x^2^ (2), 0.85, *p* = 0.009) and humpback whales ß = −4.53, SE = 1.33, Wald x^2^ (1) = 11.55, *p* = 0.007). A moderate, yet statistically significant, relationship was observed between lamprey bite marks and prevalence of pale skin patch syndrome in fin whales (*r_pb_* = 0.41, *p* = 0.002) and lamprey bite marks and the prevalence of dark focal skin diseases in humpback whales (*r_pb_* = 0.48, *p* = 0.0005). Cutaneous nodules were more prevalent in humpback whales, with 68% of individuals affected compared with only 4% in fin whales. Conversely, the predominant cutaneous lesion in fin whales was lamprey bite marks with 74% of individuals bearing these parasitical lesions, compared to 36% in humpback whales (Table 3). Uniquely observed in humpback whales, barnacle (epibiont) loads compared between the right- and left-side appendages of the fluke showed a strong symmetrical load (*r* = 0.68, *p* ˂ 0.001) and a significant relationship between congener markings and prevalence of light focal skin disease was observed (*r_pb_* = 0.44, *p* = 0.001).

One cutaneous lesion classified in the miscellaneous category of skin conditions, which we describe as a ***bulla-like lesion*** (Figure 11), was present in 16% of humpback whales and 2% of fin whales. This skin anomaly had a small, yet significant impact on the overall skin condition scores of humpbacks (ß = −0.93, SE = 0.38, Wald x^2^ (1) = 5.99, *p* = 0.001) but not in fin whales (ß = −0.17, SE = 0.35, Wald x^2^ (1) = 0.26, *p* = 0.56).

### 3.3. Environmental Parameters

Biotic parameters regarding annual zooplankton biomass and abundance of pelagic fishes did not correlate with body condition scores in humpbacks or fin whales (all *p* > 0.07). However, abiotic environmental parameters did have significant relationships with some skin diseases and parasite/epibionts in both fins and humpback whales (Table 4). Low saturated oxygen levels (DO) (*M* = 17.06, *SD* = 17.93), mostly in the Rimouski area, were correlated with high levels of nitrate (*r* = −0.49, *p* ˂ 0.001), phosphate (*r* =−0.74, *p* ˂ 0.001), chlorophyll A (*r* = −0.39, *p* = 0.006), sea temperatures (*r* = −0.52, *p* ˂ 0.001) and salinity levels (*r* = −0.56, *p* ˂ 0.001) consistent with eutrophication. A total of six cutaneous lesion categories were related to environmental parameters in one of the two species. Light focal skin disease, tortuous cutaneous disease and cookiecutter wounds were not correlated to any environmental parameters (for both species). Finally, one category (pale skin patch syndrome) was related to higher dissolved oxygen levels in both species (Table 4).

### 3.4. Aggregation of Physical Measures Inferring Overall Welfare State

Positive welfare was associated with higher salinity levels and lower sea temperatures in both species (all *p* ≤ 0.002) and poorer welfare scores were significantly related to scars and injuries originating from human activity in fin (*r* = −0.49, *p* ≤ 0.001) and humpback whales (*r* = −0.50, *p* ≤ 0.001) (Table 5).

## 4. Discussion

Welfare assessment of wild cetaceans is increasingly gaining scientific attention as a preventive approach to inform international conservation bodies about priority concerns. For instance, in 2014, the International Whaling Commission (IWC) founded a working group on emerging welfare issues, resulting in the publication of a new theoretical framework for the welfare assessment of wild cetaceans and recommending further development of this framework based on field research [10,22]. Our study present advances in validating a first physical indicator of welfare that could be included in an overall assessment protocol for humpback and fin whales found in a highly anthropized environment by analyzing over six thousand images and demonstrating positive correlations among four species-specific physical measures that can be used to partially assess welfare. Physical measures in both species clustered in two categories, one related to physical state (body condition and injuries) and the other to the epidermal state (skin condition and parasites/epibionts).

Based only on the physical measures of welfare, our results suggest that most humpback whales assessed in our study were in a positive welfare state compared to fin whales, who were in a moderate one. The main welfare issues, in both species, were specifically related to two underlying factors: the cumulative effects of eutrophication on the environment, and direct anthropogenic activities associated with fishing activities and boat collisions. We discuss our results considering these two potential welfare issues.

### 4.1. Effects of Environmental Degradation on Epidermal State

Skin diseases affecting cetaceans can be caused by several pathogens including viruses, bacteria or fungi, and have been correlated with environmental parameters such as low salinity levels and high water temperatures [100,157]. However, these two measures are also intrinsically correlated to other marine environmental parameters, notably: dissolved oxygen levels, integrated nitrate, phosphate and chlorophyll-A levels. Our study showed that, between 2016 and 2021, dissolved oxygen levels in the St-Lawrence estuary reflected hypoxia and were highly correlated with all other environmental parameters suggesting eutrophication [158,159,160]. We found that pale skin patch syndrome prevalence was observed in both species when dissolved oxygen levels were higher, suggesting a non-pathological epidermal condition like desquamation. Epidermal growth and renewal is rapid and continuous in cetaceans and actively maintains a protective barrier from environmental stressors [161]. Furthermore, epidermal homeostasis disruption has been linked to systemic imbalance of nutrient levels, like iron, in aquatic and terrestrial animals, underlying the importance of skin health on overall welfare [162,163]. Conversely, lower dissolved oxygen levels were correlated to the prevalence of tattoo skin disease-like lesions in fin whales, suggesting that this environmental parameter might have greater impact on cetacean skin health than previously thought. This hypothesis should be further explored.

Among cutaneous conditions, light focal skin disease had the highest prevalence in both species and was not correlated to any environmental parameters, suggesting that this cutaneous condition might have an infectious etiology, as shown in previous studies [139,140,164]. Parasite and epibiont loads were significantly different between the two species, where lamprey bite marks, cookiecutter wounds and orange film (diatoms) affected more fin whales than humpbacks. Sea lice and barnacles were only observed in humpback whales and the latter was the highest epibiont prevalence in this species.

Recent studies on the degradation of the marine environment due to climate change have highlighted some relationships between abiotic changes (higher sea temperature, lower salinity levels, etc.) and the proliferation of some parasites, like sea lampreys [165,166]. Furthermore, higher sea temperatures seem to increase the survival of some pathogens that could negatively impact cetaceans’ welfare, but more studies on this potential problem are needed [167,168].

### 4.2. Anthropogenic Activities and Stress Response

The strongest finding of our study was the significant relationship between lesions from anthropogenic origins and thinner body conditions, even when animals had clearly healed wounds characterized by superficial scars. This relationship was observed in both species despite different prevalences of anthropogenic injuries among them. Moreover, no significant relationships were seen between body condition and prey availability. This finding highlights the long-term effects of severe anthropogenic injuries and, besides direct physical impairment, may suggest a chronic stress response impacting their overall welfare.

In free-ranging baleen whales, chronic stress can occur following various natural and anthropogenic impacts, particularly if they are prolonged, severe, unpredictable and/or occur simultaneously with other stressors. For example, entanglement in fishing gear seems to be the greatest source of chronic stress in North Atlantic right whales [169,170,171]. Fecal glucocorticoid metabolite concentrations have been found to be much higher in free-ranging chronically entangled whales than in non-entangled ones [169,171]. Moreover, postmortem steroid hormones extracted from the baleen plates of an NARW were positively correlated with the period of entanglement (the dates the animal was visually observed in the field) and showed elevated levels above baseline [170]. Similarly, an 11-year-old North Atlantic humpback whale who suffered chronic entanglements during her lifetime had deep chronic skeletal lacerations and higher corticosterone levels than three other specimens who died from diseases or ship strike [71]. Long-term studies of rorqual whales observed in the Gulf of St-Lawrence suggest a gradual decline in abundance and survival since early 2000 in fin whales [172], and a decline in reproductive success in humpback whales [173]. Perhaps not coincidently, entanglement rates of Northwest Atlantic baleen whales have been sharply rising since the mid-1990s, due to changes in fishing rope material, comprised now of higher-strength polymer fibres than previously [174]. Consequently, whale entanglements are more severe, entangled whales are less likely to free themselves, and even if whales shed the gear and survive, they may have long-lasting injuries and chronic stress [92,170]. This negatively affects fitness and the immune response through the continued release of glucocorticoids and increases the risk of infectious diseases [175,176]. Our study further confirms that acute and chronic entanglements in fishing gears represent a serious threat to the welfare of North Atlantic whales.

### 4.3. Strengths and Limitations

This study is the first to validate a physical indicator of welfare in free-range mysticetes like fin and humpback whales by correlating four animal-based physical measures. Our scoring system was proven to be reliable and valid, especially by weighting the scores using the CRITIC methodology. Furthermore, we showed convergent validity between body condition and injury scores, and between skin condition and parasite/epibiont scores, in both humpback and fin whales. Finally, this study showed significant statistical relationships between parasitical prevalence and skin diseases, and that whales injured by anthropogenic activities were thinner than the whales who had no evidence of past/present injuries. However, sex, personality and early life experiences can impact the underlying stress response mechanisms and overall welfare of each animal, but these could not be taken into account, highlighting the limitations of our study. Furthermore, the impact of environmental variables on the welfare status of animals is difficult to measure on free-ranging whales navigating complex marine ecosystems. Finally, our physical indicator can only partially assess welfare since it needs to correlate with behavioural and physiological indicators to validate the overall welfare status of an animal. The inevitable subjective aspect of our measuring scales’ thresholds (negative/moderate/good) can only be objectively validated over time when correlated with other measures.

## 5. Conclusions

The purpose of this study was to statistically validate a physical indicator of welfare for humpback and fin whales living in an anthropized environment. Based on a multi-scale scoring system of body, skin, injury and parasite/epibiont condition measures, our results showed positive inter-correlation and discrimination between all measures validating this indicator. Overall welfare states, for both humpback and fin whales, were mostly impacted by the degradation of the marine environment and previous physical trauma due to anthropogenic activities. Although the majority of humpback whales were found to be in an overall good welfare state whilst fin whales were in a moderate one, the true potential of this first indicator to be included in a welfare assessment protocol will only be fully measured through long-term studies. Future research should focus on validating the current four measures of physical state in other populations of humpback and fin whales, as well as in other species like the grey whales, who are showing signs of chronic stress and poor health. Finally, the measures included in our physical indicator of welfare need to be correlated with other measures/indicators like physiological measures of stress and behavioural responses.

## Figures and Tables

**Figure 1 animals-14-03519-f001:**
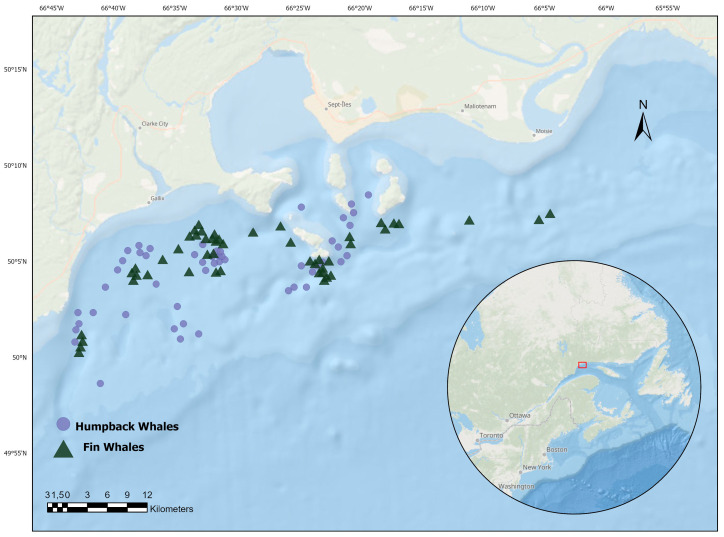
Research area in the region of Sept-Îles and Port-Cartier, showing humpbacks (n = 50) represented by circles and fin whales (n = 50) by triangles, observed as part of the current study and between 2016–2021.

**Figure 2 animals-14-03519-f002:**
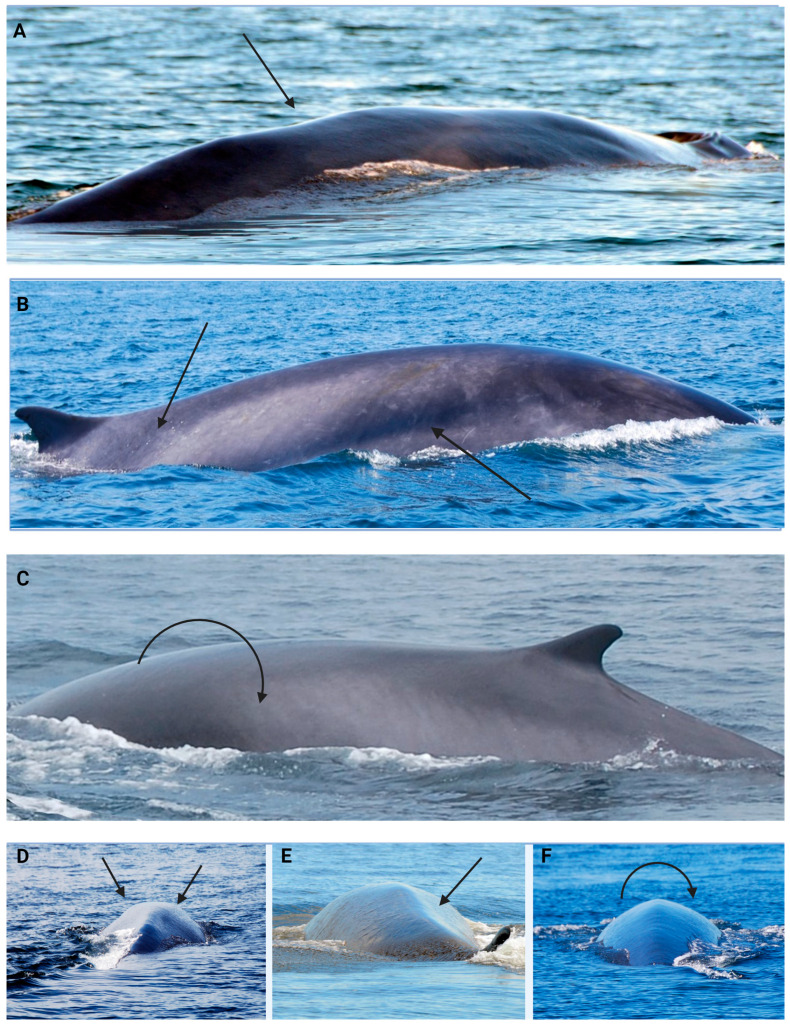
Body condition scoring system with top horizontal photographs showing a lateral view and bottom square photographs showing a posterior view. Photos (**A**,**D**) represent a score of 0–1, considered emaciated since we can see the dorsal vertebra and ribcage. Photos (**B**,**E**) represent a score of 2–3 considered thin/moderate since we can distinguish a prominent dorsal ridge with sunken flanks. Photos (**C**,**F**) represent an optimal score of 4 where the curved arrows show no ridge and a fully rounded body.

**Figure 3 animals-14-03519-f003:**
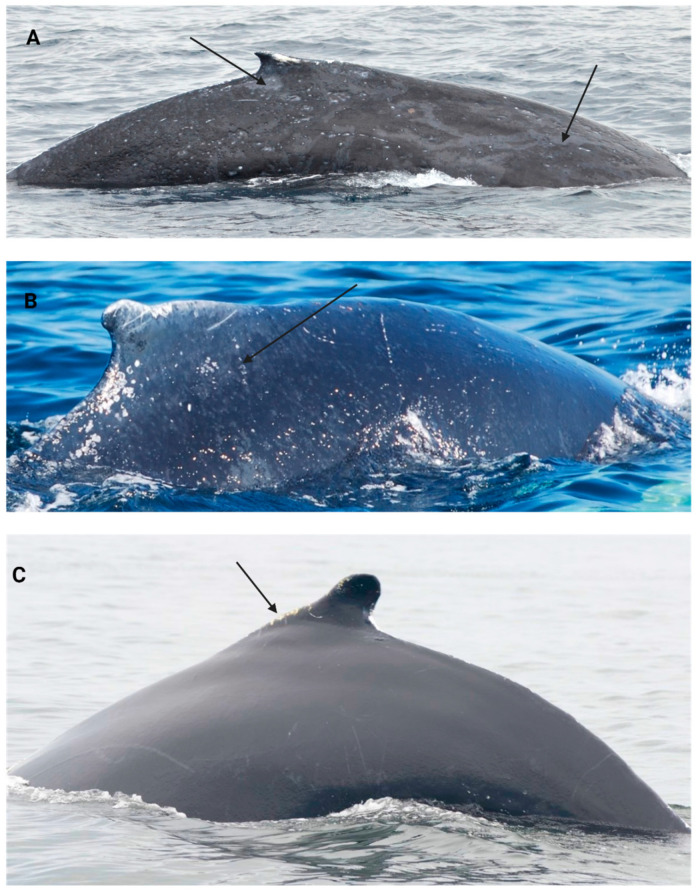
The skin condition scale used in humpback whales. (**A**) represents a skin condition of 1, where the arrows point to one skin lesion type covering ≥ 75% of the observable body surface. (**B**) represents a score of 2, where the arrow points to one type of skin lesion covering ≥ 25% and ≤74% of the observable body surface. (**C**) represents a skin condition score of 4, where the arrow points to ≤5% of skin lesions on the observable body surface.

**Figure 4 animals-14-03519-f004:**
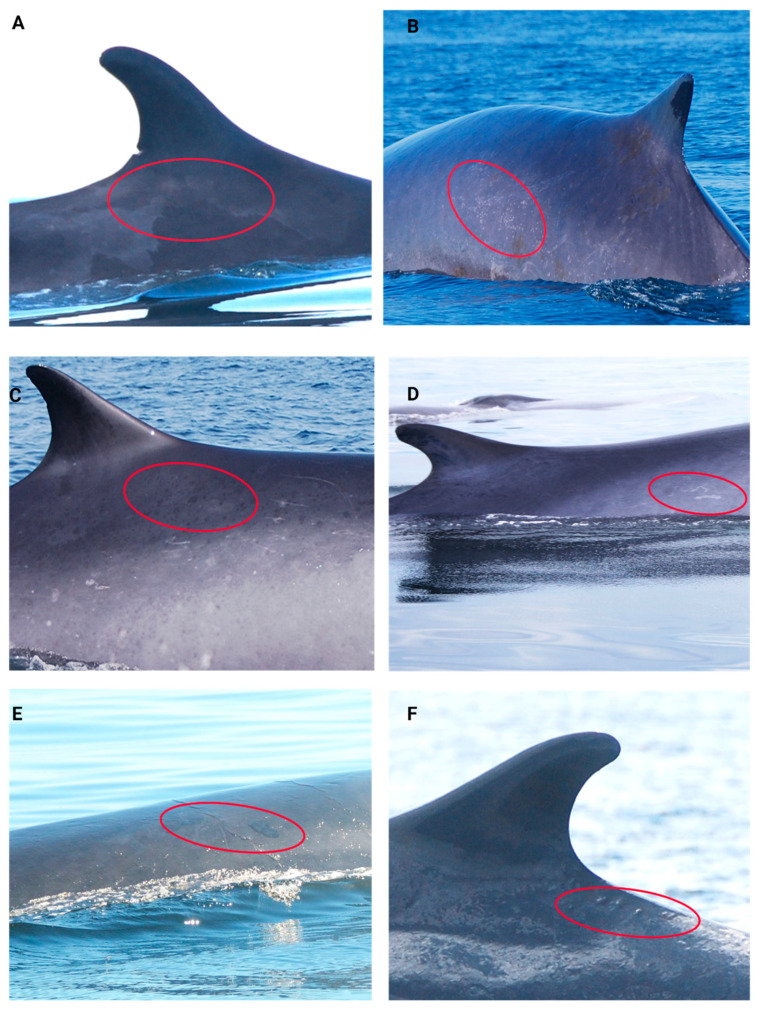
Cutaneous lesions categories used in the scoring of skin condition in fin whales (observed in the red circles). (**A**) Pale skin patch syndrome; (**B**) light focal skin disease; (**C**) dark focal skin disease; (**D**) tortuous cutaneous marks; (**E**) tattoo skin disease-like lesions; (**F**) cutaneous nodules.

**Figure 5 animals-14-03519-f005:**
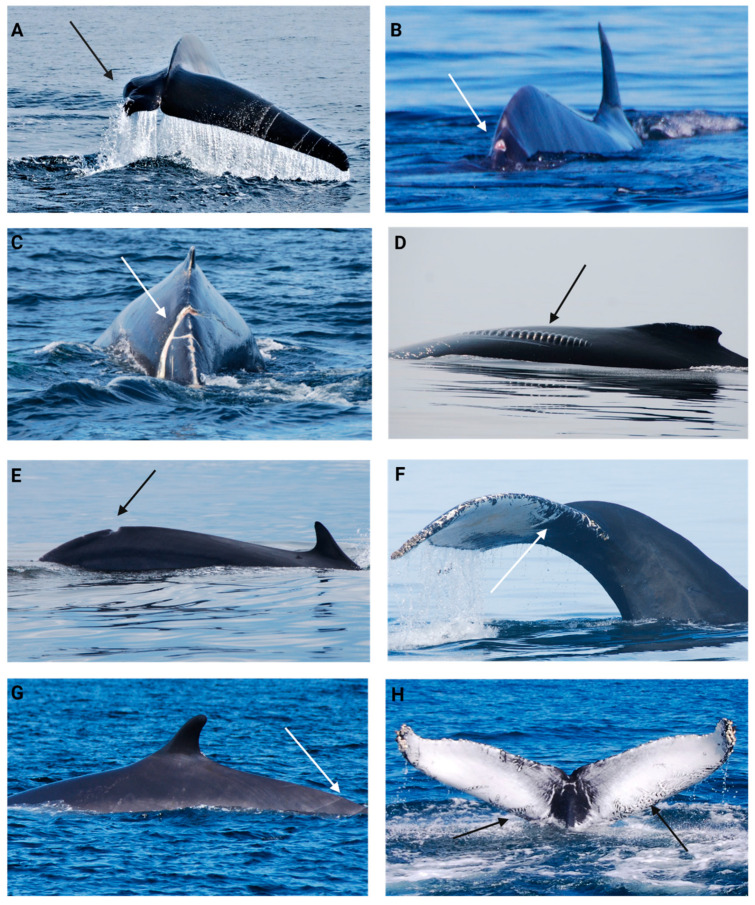
Injury condition scoring system. (**A**) A score of 0 for a fin whale missing half its fluke. (**B**) A score of 1 for a fin whale with open wounds on the peduncle. (**C**) A score of 2 for a fin whale with deep scars typical of entanglement in fishing gear. (**D**) A score of 2 for a humpback with an important scar from a boat propeller. (**E**) A score of 2 for a deep entanglement scar in a fin whale. (**F**) A score of 3 for superficial entanglement scars in a humpback. (**G**) A score of 3 for superficial entanglement scars in a fin whale. (**H**) A score of 3 for important predation tooth rake on a humpback.

**Figure 6 animals-14-03519-f006:**
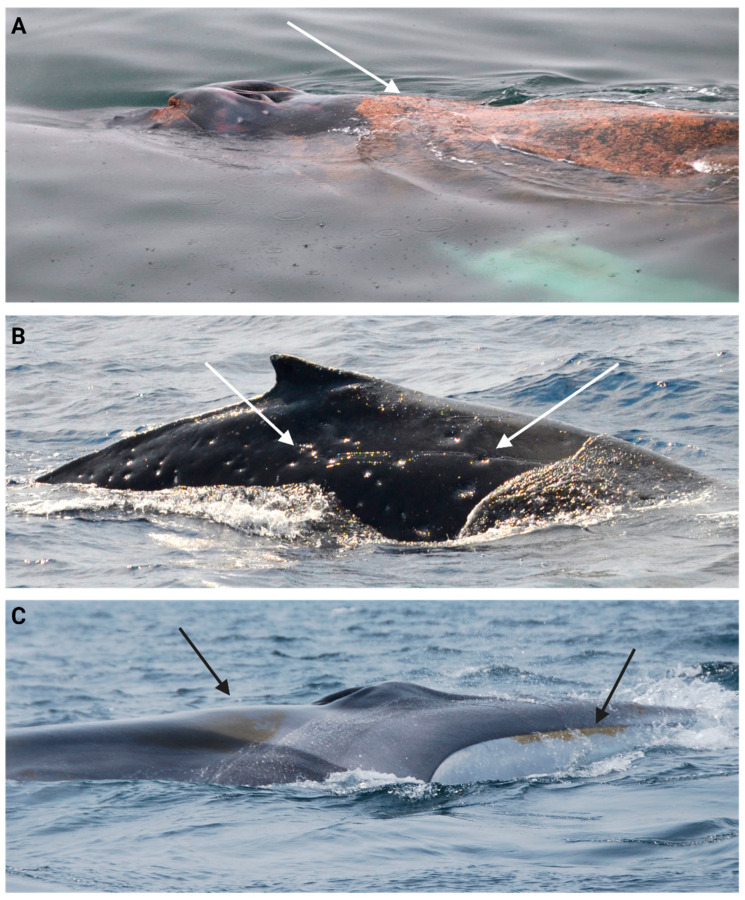
Parasite and epibiont condition scoring scale where (**A**) a score of 1 with ectoparasite (sea lice) covering ≥ 40% of observed body; (**B**) a humpback whale with a score of 2 with cookiecutter scars covering ≥ 5% and ≤40%, and (**C**) a fin whale scored as 3, with a small patch of diatoms covering ˂ 5% of its body.

**Figure 7 animals-14-03519-f007:**
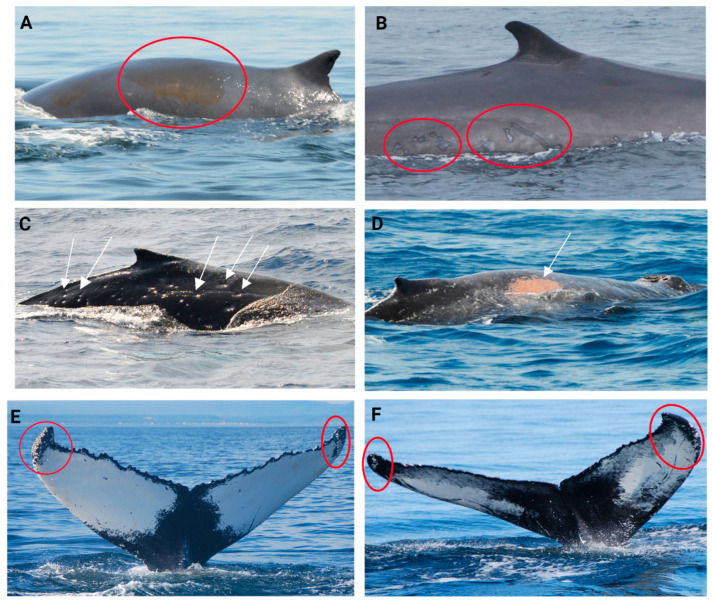
Parasite and epibiont categories: (**A**) fin whale with an orange film (diatoms); (**B**) fin whale with lamprey bite scars and skid marks; (**C**) humpback whale with multiple cookiecutter scars; (**D**) humpback whale with whale lice infestation on an apparent healing wound; (**E**) humpback whale fluke with barnacles covering the tip of the left- and right-side fluke lobes in a symmetrical load; (**F**) humpback whale fluke without barnacles on tips of the left- and right-side fluke lobes, also in a symmetrical manner (highlighted by the red circles and arrows).

**Figure 8 animals-14-03519-f008:**
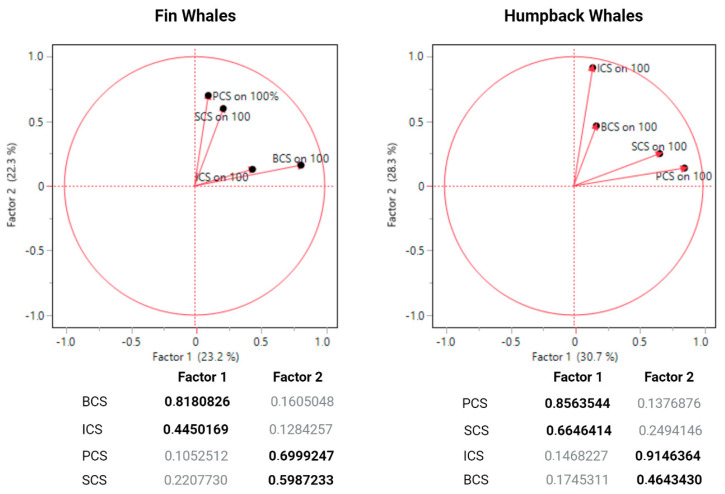
Underlying factors for physical condition measures: body condition score (BCS), injury condition score (ICS), parasites/epibionts condition score (PCS) and skin condition score (SCS) in fin whales (n = 50) and humpback whales (n = 50).

**Figure 9 animals-14-03519-f009:**
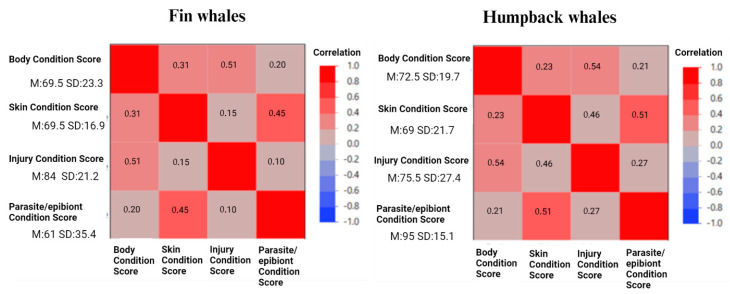
All four measuring scales positively correlated in both species (n = 50 fin whales and n = 50 humpback whales), and no correlations were stronger than 0.75, validating the relevance of including each measure in our physical welfare indicator (discriminant validity). M = mean and SD = standard deviation. All values were statistically significant (*p* ≤ 0.05).

**Figure 10 animals-14-03519-f010:**
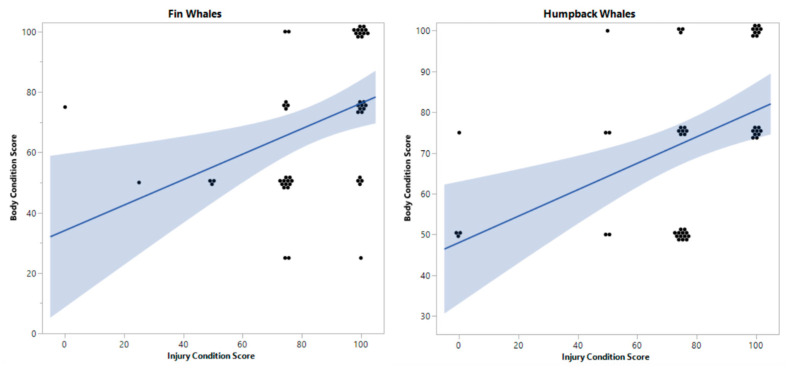
Ordinal regression showed that injury conditions from anthropogenic origins, related positively to body condition in both fins and humpback whales; linear fit for fin whales (F (1,48) = 8.34; *p* = 0.005) and humpback whales (F (1,48) = 12.21; *p* = 0.001).

**Figure 11 animals-14-03519-f011:**
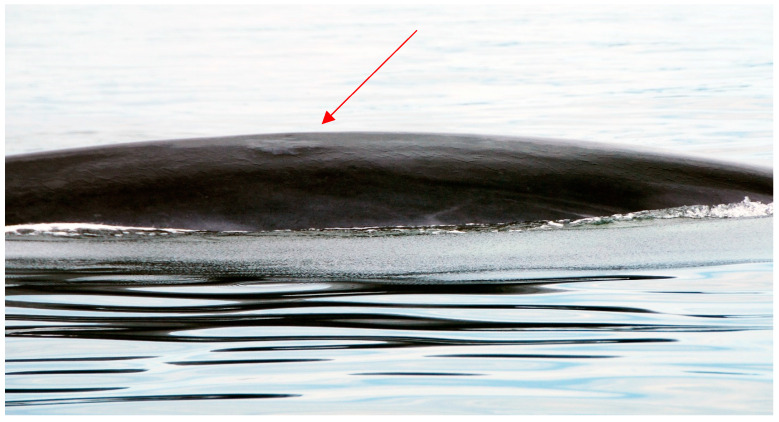
In the miscellaneous category of cutaneous lesions associated with skin diseases (see Table 2 in Section 2.4.2), we included an undocumented condition characterized by a raised fluid-filled epidermis lesion, which we categorized as ***Bulla-like lesion***.

**Table 1 animals-14-03519-t001:** The physical indicator of welfare scoring system includes a 0–4 scale for body condition, skin condition, injury and scarring condition measures, and a 1–3 scale for parasite and epibiont condition scores. The score of each measure was then transformed to a percentage scale and weighted according to the CRITIC method. The final score for each whale was then associated with the corresponding inferred welfare state.

Welfare Measures (Criteria)	Scoring Scale	Inter-Criteria Correlation Weight
Fin Whales	Humpbacks
Body Condition Score	0-1-2-3-4	23.71%	25.24%
Skin Condition Score	0-1-2-3-4	18.26%	24.08%
Injury/Scarring Condition Score	0-1-2-3-4	24.46%	32.91%
Parasite/Epibiont Condition Score	1-2-3	33.57%	17.76%
	**Total =**	**Weighted scores**
Physical Indicator Score	**Inferred welfare state**
	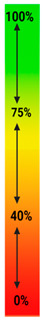	Physical condition is optimal, suggesting comfort of good health, high functional capacity and postprandial satiety, inferring **positive welfare state**.Physical condition is moderate, suggesting feelings of hunger and/or psychophysical exertion, inferring **moderate welfare state**.Physical condition is poor, suggesting chronic pain due to hunger, psychophysical exertion and underlying pathological states, inferring **negative welfare state**.

**Table 2 animals-14-03519-t002:** Characteristics of cutaneous lesions categories associated with skin diseases used for the skin condition scoring.

Skin Lesions Categories	Description
Pale skin patch syndrome (PSP)	Areas of opaque to translucent skin, light grey or whitish colouration [139] (Figure 4A).
Light focal skin disease (LFD)	Clusters of distinct, smallish round or oval white/light grey lesions [139] (Figure 4B).
Dark focal skin disease (DFD)	Clusters of distinct, smallish round or oval black/dark grey lesions [139] (Figure 4C).
Tortuous cutaneous marks (TCM)	Black or white linear lesions leaving tortuous tracks, with raised or depressed patterns [140] (Figure 4D).
Tattoo skin disease-like lesions (TSD-L)	Dark or light grey rounded borders with a characteristic stippled pattern [7] (Figure 4E).
Cutaneous nodules (NOD)	Circumscribed nodules with grey or normal pigmentation [68] (Figure 4F).
Miscellaneous	Other cutaneous lesions not associated with current categories.

**Table 4 animals-14-03519-t004:** Point-biserial correlation coefficients and associated *p* values between abiotic parameters (sea temperature (TC°) and salinity levels (S%); dissolved oxygen (DO); integrated mean of nitrate (NO_3_^−^); phosphate (PO_4_^3−^); silicate (O_3_SI^−2^) and cutaneous lesion types in fin and humpback whales: pale skin patch syndrome (PSP); light focal skin disease (LFD); dark focal skin disease; cutaneous nodules (NOD); Bulla-like lesions (BLLs); tortuous cutaneous mark (TCM); tattoo skin disease- like lesions (TSD-Ls; only in fin whales); cookiecutter shark wounds (CCWs); sea lamprey bite and skidding marks (LBMs); prange film diatoms (OFD); and, only found in humpbacks, sea lice (SL) and barnacles (BRs). The prevalence of cutaneous lesions for each species is presented in % and statistically significant values in bold.

Fin Whales													
Lesion Correlation Coefficient and *p* Values Prevalence													
	TC°	*p*	S %	*p*	DO	*p*	NO_3_^−^	*p*	PO_4_^3−^	*p*	O_3_SI^−2^	*p*	%
PSP	0.30	**0.030**	−0.31	**0.002**	0.36	**0.009**	0.01	0.944	−0.27	**0.004**	0.00	0.999	0.04
LFD	0.04	0.789	0.08	0.568	−0.01	0.906	0.07	0.602	0.23	0.103	0.02	0.883	0.56
DFD	−0.35	**0.014**	−0.35	**0.012**	0.19	0.173	−0.32	**0.025**	−0.37	**0.008**	0.07	0.594	0.32
NOD	−0.42	**0.002**	−0.50	**0.000**	0.39	**0.005**	0.04	0.757	−0.40	**0.000**	0.18	0.203	0.04
BLL	−0.17	0.229	−0.20	0.149	0.17	0.233	−0.06	0.652	−0.19	0.172	0.02	0.874	0.02
TCM	−0.05	0.730	0.02	0.864	−0.11	0.419	0.12	0.372	0.18	0.201	0.11	0.441	0.20
TSD	0.09	0.501	0.10	0.471	−0.30	**0.031**	−0.06	0.666	−0.02	0.857	−0.14	0.406	0.08
CCW	−0.09	0.509	−0.06	0.656	0.06	0.656	0.06	0.632	0.08	0.549	0.09	0.500	0.72
LBM	−0.09	0.514	−0.11	0.422	0.13	0.337	0.07	0.601	0.03	0.824	0.06	0.632	0.74
OFD	−0.18	0.188	−0.20	0.160	0.38	**0.007**	0.19	0.164	−0.01	0.906	0.14	0.321	0.54
**Humpback whales**													
**Lesion Correlation Coefficient and *p* values Prevalence**													
	**TC°**	** *p* **	**S %**	** *p* **	**DO**	** *p* **	**NO^−^_3_**	** *p* **	**PO^3−^_4_**	** *p* **	**O_3_SI^−2^**	** *p* **	**%**
PSP	−0.14	0.317	−0.19	0.182	0.29	**0.041**	0.17	0.222	−0.12	0.368	0.16	0.263	0.56
LFD	0.08	0.571	0.06	0.631	0.14	0.321	0.32	0.063	0.02	0.868	0.16	0.241	0.48
DFD	0.08	0.535	0.01	0.893	0.07	0.621	0.05	0.679	−0.12	0.370	−0.04	0.739	0.16
NOD	−0.19	0.662	−0.19	0.182	0.18	0.193	−0.06	0.667	−0.08	0.578	0.02	0.858	0.68
BLL	−0.09	0.526	−0.16	0.265	0.02	0.852	−0.37	**0.008**	−0.12	0.377	−0.29	**0.044**	0.16
TCM	0.01	0.971	0.09	0.528	−0.02	0.889	0.22	0.120	0.20	0.158	0.19	0.180	0.02
CCW	0.24	0.091	0.26	0.063	−0.14	0.317	0.27	0.054	0.14	0.328	0.15	0.284	0.24
LBM	0.33	**0.019**	−0.08	0.560	0.18	0.203	−0.38	**0.006**	−0.08	0.540	−0.25	0.075	0.36
OFD	−0.22	0.120	−0.24	0.083	0.25	0.075	−0.02	0.845	−0.10	0.470	0.01	0.972	0.24
SL	0.33	**0.018**	−0.32	**0.025**	0.20	0.152	−0.01	0.968	−0.32	**0.021**	−0.08	0.549	0.40
BR	−0.31	**0.026**	0.038	**0.005**	−0.25	0.069	−0.02	0.873	0.34	**0.015**	0.03	0.810	0.84

**Table 5 animals-14-03519-t005:** Mean welfare scores (weighted scores on a 0–100 scale): body condition score (BCS), skin condition score (SCS), injury condition score (ICS), parasite/epibiont score and overall welfare score (OWS).

Species	BCS	SCS	ICS	PCS	OWS
*M* (*SD*)					
Bp	69.05 (23.30)	69.5 (16.97)	84.0 (21.28)	61.0 (35.41)	70.18 (17.93)
Mn	72.5 (19.72)	69.2 (21.75)	75.5 (27.42)	95.2 (15.15)	76.63 (16.10)

## Data Availability

The original contributions presented in the study are included in the article, further inquiries can be directed to the corresponding author.

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
