# Peer review of "Physical Measures of Welfare in Fin (*Balaenoptera physalus*) and Humpback Whales (*Megaptera novangliae*) Found in an Anthropized Environment: Validation of a First Animal-Based Indicator in Mysticetes"

_animals, 2024, doi:10.3390/ani14233519_

Round 1

Reviewer 1 Report

Comments and Suggestions for Authors

This is an excellent and important paper. It reports a new approach to physical assessment of wild cetaceans and I agree wholeheartedly with the authors that this is important in terms of its implications for cetacean conservation as well as welfare. The fact that it is based on analyses of more than 6400 images is remarkable and underpins its importance.

I have just a few small suggestions that I believe may improve the paper.

Firstly, I find the images in figure 4 – the cutaneous lesions – difficult to see. I suggest that you review these and if you have any better photos of these lesions, consider substituting them. There may be a balance between showing them in close up so that their details can be seen and showing the body of the whale. However, as you are not trying in these images to show what proportion of the body is affected, close ups of the lesions may be preferable.

Secondly, the term ‘validation’ or ‘validates’ is used as one of the main claims in this paper. As, for example, in the abstract where it states ‘This study validates a first evidence-based physical health and welfare assessment protocol for...’ etc.

However, isn’t this paper, in fact, actually the very first to ever present or provide an assessment based on measurements made in the field of observed factors seen in wild cetaceans? That would seem to me to be its most important claim. In my head at least any 'validation' of this approach is then a further step. I don’t want to get hung up on semantics, but I would encourage you to consider this as you might inadvertently be ‘underselling’ the paper.

Thirdly, at line 57-59, the following is stated: ‘In this regard, legislators have been urged to include cetacean welfare assessment protocols in regulations and conservation plans, notably by the Committee on Welfare Issues of the  International Whaling Commission [8–10,19–21]’.

I think that the general thrust of what is being stated in this paragraph is correct. There has been a growing interest in the welfare of wild cetaceans and there have been calls for this to be added to conservation efforts. However, the relevant IWC body is I think called the ‘Whale Killing Methods and Welfare Issues Working Group’ and its report was not authored by Butterworth as you have it in the reference list. You can find the report here: https://archive.iwc.int/pages/view.php?ref=6230&k=

Please look at sources and the sentiments expressed here carefully.

Fourthly, at line 198 you say ‘If an animal actively avoided the boat, the focal study was terminated.’ It would probably be of interest and assistance to the reader to know a little more about how this judgement was made – i.e. what was the observed behaviour that led to the conclusion that the boat was being avoided.

Author Response

General comment:

This is an excellent and important paper. lt reports a new approach to physical assessment of wild cetaceans and I agree wholeheartedly with the authors that this is important in terms of its implications for cetacean conservation as well as welfare. The fact that it is based on analyses of more than 6400 images is remarkable and underpins its importance.

General response: Thank you for this positive comment which recognizes the relevance of our study and the underlying amount of data analysis we have undertaken to produce these results. All co-authors are extremely grateful for this comment. Thank you, it is much appreciated!

Comment 1: Firstly, I find the images in figure 4 - the cutaneous lesions - difficult to see. I suggest that you review these and if you have any better photos of these lesions, consider substituting them. There may be a balance between showing them in close up so that their details can be seen and showing the body of the whale. However, as you are not trying in these images to show what proportion of the body is affected, close ups of the lesions may be preferable.

Response 1: Thank you for this comment and we totally agree with you. When we look at previous published articles in the Journal Animals, we have noticed that some, if not most, photographs or important Figures are outlined in the text as “pop up images” which allows the reader to click on it and have a much better view of the image/Figure. 

If the editors of Animals edit our manuscript to include such “popups” this is what it would look like wen clicking on Figure 4 (we created a wee video for you): https://www.youtube.com/watch?v=PWszIc3dP_I

Comment 2:  Secondly, the term 'validation' or 'validates' is used as one of the main claims in this paper. As, for example, in the abstract where it states This study validates a first evidence-based physical health and welfare assessment protocol for…  etc. However, isn't this paper, in fact, actually the very first to ever present or provide an assessment based on measurements made in the field of observed factors seen in wild cetaceans? That would seem to me to be its most important claim. ln my head at least any 'validation' of this approach is then a further step. I don't want to get hung up on semantics, but I would encourage you to consider this as you might inadvertently be underselling' the paper.

Response 2: Thank you for this important comment. The use of the term Validation is extremely important because we are referring to the actual statistical validation of this “first” evidence-based physical state and welfare assessment protocol conducted in wild cetaceans. This is why we have used this specific word, although we agree with you that this is not evident for all readers. We have rephrased our title:

Physical Measures of Welfare in Fin (Balaenoptera physalus) and Humpback Whales (Megaptera novangliae) found in an Anthropized Environment: Validation of a first Animal-Based Indicator in Wild Cetaceans (Line 2-5)

and our conclusion: “The purpose of this study was to statistically validate a first indicator of welfare for humpback and fin whales living in an anthropized environment. Based on a multi-scale” (Line: 579)

We also specified this aspect, mostly in the methodology and result sections:

- “For repeatability and validation purposes [73,74,121], a single observer”. (Line 208-209).

- “We hypothesized that the four different animal-based measures — body condition, injury condition, skin condition and parasite/epibionts condition — would show convergent validity, supported by these measures loading onto a shared factor in common factor analysis. Additionally, we theorized that these measures would show discriminant validity, supported by only low to moderate positive correlations between them. Furthermore, we assumed that normal environmental indices (lower water temperature, higher salinity levels, higher oxygen and lower nitrate levels) would correlate with good skin health and lower parasite/epibionts loads, and prey availability would correlate positively with body condition scores”. (Line: 175-183)

- Discriminant validity between the four measurement scales (body, skin, injury and parasite/epibiont) of our physical welfare assessment protocol was tested with Spearman correlations within each species. Discriminant validity is supported if correlations between two scales are less than 0.75 [149]. We performed a common factor analysis (Q-type) using Maximum-Likelihood and Varimax orthogonal rotation to test convergent validity between the measurement scales of humpbacks and fin whales”. (Line: 359-365)

- “Furthermore, we showed convergent validity between body condition and injury scores, and between skin condition and parasite/epibiont scores, in both humpback and fin whales”(Line: 573-575)

Comment 3:  Thirdly, at line 57-59, the following is stated: 'ln this regard, legislators have been urged to include cetacean welfare assessment protocols in regulations and conservation plans, notably by the Committee on Welfare Issues of the International Whaling Commission [8- 10, 19-21)'. I think that the general thrust of what is being stated in this paragraph is correct. There has been a growing interest in the welfare of wild cetaceans and there have been calls for this to be added to conservation efforts. However, the relevant IWC body is I think called the 'Whale Killing Methods and Welfare Issues Working Group' and its report was not authored by Butterworth as you have it in the reference list. You can find the report here: https://archive. iwc.int/pages/view.php?ref=6230&k= Please look at sources and the sentiments expressed here carefully.

Response 3: Thank you for pointing out the correct working group, we have changed it: “In this regard, legislators have been urged to include cetacean welfare assessment protocols in regulations and conservation plans, notably by the Whale Killing Methods and Welfare Issues working group of the International Whaling Commission [8–10,20–22]”. (Line 58-59)

However, I’m really talking about Butterworth’s report.

https://scholar.google.ca/scholar?hl=fr&as_sdt=0%2C5&q=Butterworth%2C+A.+Report+of+the+Workshop+to+Support+the+IWC%E2%80%99s+Consideration+of+Non-Hunting+Related+Aspects+of+Cetacean+Welfare%2C+Kruger%2C+S+Africa%2C+2+to+6+May+2016%3B+Proceedings+International+Whaling+Commission%3A+Cam-bridge%2C+2017%3B+Vol.+IWC+BT&btnG=

Comment 4, Fourthly, at line 198 you say 'If an animal actively avoided the boat, the focal study was terminated.' lt would probably be of interest and assistance to the reader to know a little more about how this judgement was made - i.e. what was the observed behaviour that led to the conclusion that the boat was being avoided.

Response 4: Thank you for your question. There are sometimes subtle behaviours that alarm us that the animal might be stressed by the boat approaching (sometimes at more than 300 meters), like changing direction or spouting heavily, but at such distance, it might also be a normal behaviour. What we do then, is slow down the boat and observe the subsequent behavioural responses. The clear sign of stress is when we get closer, at around 100 meters, the animal will instantly dive. However, we specified further here: 

The maximum time spent with an individual was two hours, including diving and breathing sequences. If an animal actively avoided the boat (e.g. dives immediately when the boat approaches), the focal study was terminated”. (Line 201-202)

Reviewer 2 Report

Comments and Suggestions for Authors

Please see the comments on the word document

Comments on the Quality of English Language

Minor English grammar requires improvement

Author Response

General comment:

The current manuscript is an important addition to our marine mammal community. However, while I am pleased that the authors are trying to add to the limited welfare related literature on cetaceans, this paper is not a welfare assessment, but instead is assessing some health indicators that could be used in the future to form part of a welfare assessment. The title and focus of the paper must be altered to this focus. This paper should instead of arbitrarily trying to assess welfare poorly, provide a robust assessment of some health indicators in these species and potential environmental causes. The discussion and conclusion could then be utilised to highlight how this improved knowledge on health and environmental parameters could be used in the future to inform the development of a welfare assessment framework specic to these species. Often had many difficulties in being accepted, due to many scientists misunderstanding the ‘ease’ of assessing and inferring welfare states. In its current format this paper would only further this line of thought and could hinder rather than help the integration of animal welfare science with conservation biology. Additionally, I suggest that the authors delve deeper into recent literature, particularly as a number of papers on cetacean welfare are missing (See suggestions for reading in reference section at end of this document). They must carefully consider the tone and wording of the paper so as not to over- step and suggest that their results can be used to infer more than is really possible. To ensure a strong, robust animal welfare science foundation is established within the marine mammal community we must carefully explain what and how welfare science can be applied and the various caveats that must be considered.

General response: Thank you for taking the time to review our manuscript, we agree with most of your comments and will address each specific one below. We also want to thank you for sharing some interesting references, especially Miller et al. 2018, which we had forgotten about and is relevant to our manuscript. We will reference this article in some of our comments.

Comment 1 Title: The title of the paper is misleading, as it is not the first paper to provide a welfare assessment of wild cetaceans, nor is it validating a welfare assessment protocol, since it only considers one aspect of welfare: health. Additionally, health is generally considered to be one facet of welfare, so it should not be health and welfare, instead the health parameters may be used to infer potential welfare states. Please re-write the title to focus on what the paper actually presents, which is indicators of health and use welfare as one of the key words instead. This part of the title must be deleted as it is incorrect “Validation of a First Animal-Based Welfare Assessment Protocol for Wild Cetaceans.”

Response 1, Title: Thank you for your comment, and you are right to point out that health is just one aspect of welfare. We have changed the title to point out that we are presenting the physical measures of welfare in fins and humpback whales in regard to the animal-based physical indicator we have validated (which includes 4 measures, detailed in the manuscript):

Physical Measures of Welfare in Fin (Balaenoptera physalus) and Humpback Whales (Megaptera novangliae) found in an Anthropized Environment: Validation of a first Animal-Based Indicator in Wild Cetaceans. (Line 2-5)

Comment 2, Abstract: Line 30: What is meant by validated here? Is this in terms of it being a valid welfare indicator i.e. there is multiple measures of different types that have been assessed and provide a correlation that can be used to infer a particular welfare state, i.e. valid welfare indicator. Or is this meaning reliability in terms of the ability to assess the images and to come to similar scores among assessors?

Response 2, Abstract: Thank you for your question. We validated our physical indicator by statistically testing the measures comprised in our “physical indicator” (the terminology, physical indicator, relates to a general category, as Miller et.al.2018 defines, which subsequently includes selected measures, which we have identified based on the species-specific of fin and humpback whales’ visual features: body condition, skin condition, injury condition and parasite/epibiont condition. We therefore validated these four measures by performing, notably, a reliability and positive discrimination statistical tests. 

Comment 3, Abstract:  Line 32: Contemporary understanding of animal welfare is that welfare is the combination of physical, behavioural and physiological states and the impacts that these have on an animals mental state. Therefore, the use of ‘good physical welfare state’ is incorrect. I believe the authors mean a good physical health state. Particularly given that they consider welfare to the subjective affective state.

Response 3, Abstract: Thank you for pointing this out and you are right, as you outlined previously in comment 1. Your mention of the word state, in regard to physical, behavioural and physiological states, is what we termed as an indicator in our study (Miller et al. 2018) since these umbrella categories include a variety of species-specific selected measures (which we detail further in the manuscript) so we changed the paragraph to reflect your comment:

Anthropogenic activities impacting marine environments are internationally recognized as welfare issues for wild cetaceans. This study validates a first evidence-based physical indicator for the welfare assessment protocol of humpback (n=50) and fin whales (n=50) living in a highly anthropized environment. Visual assessments of body condition, skin health, prevalence of injuries and parasite/epibiont loads were performed using a species-specific multi-scale measuring tool. A total of 6403 images were analyzed (fin, n =3152; humpback, n= 3251) and results were validated through reliability and positive discrimination statistical tests. Based on physical measures, welfare assessment results showed that 60% of humpback whales were considered in a good welfare state compared to only 46% of fin whales. Significant relationships were observed in both species, between environmental parameters, like dissolved oxygen levels, and prevalence of cutaneous lesions like pale skin patch syndrome. Furthermore, animals with injuries due to anthropogenic activities were more likely to be in poorer body condition, suggesting chronic stress affecting welfare”. (Line 24-35).

Comment 4, Abstract:  Lines 33-34: What were the significant relationships?

Response 4, Abstract: Thank you for your question, the relationships are stated just after:   Significant relationships were observed in both species, between environmental parameters, like dissolved oxygen levels, and prevalence of cutaneous lesions like pale skin patch syndrome.

Comment 5, Abstract: Line 36: Health and welfare are not separate, health is one facet of welfare.

Response 5 Abstract: Thank you for your comment, we removed physical health from the sentence and elsewhere in the manuscript to focus only on the welfare component, since it includes health, as you pointed out in your previous comment 1:

- “Simple Summary: The welfare of free-ranging cetaceans is being impacted around the world due to human activities like commercial fishing and marine traffic. Here we validate a first non-invasive physical indicator of welfare for humpback and fin whales found in the Gulf of St-Lawrence, Canada”. (Line 16-18)

- “This study validates a first evidence-based physical indicator for the welfare assessment protocol of humpback (n=50) and fin whales (n=50) living in a highly anthropized environment” (Line 25-26)

- “Furthermore, animals with injuries due to anthropogenic activities were more likely to be in poorer body condition, suggesting chronic stress affecting welfare”. (Line 35)

- “The first indicator to be validated in a welfare protocol is usually the physical status of an animal. Depending on the species and the context, the protocol may include a wide range of measures such as body condition score [43–45], lameness score [46,47], number and severity of injuries [48,49] and clinical observation of diseases [50,51]. Once the selected measures comprised in a physical indicator are validated statistically, they can be used to infer the underlying affective state through a scoring system [41] On its own, the physical indicator can partially inform on the overall welfare of an animal but needs to be integrated into a complete species-specific welfare assessment protocol .(Line 79-86)

- “Therefore, the development of a standardized method to assess the physical state and welfare of more species of free-ranging cetaceans is needed. The aim of the current study was to validate a physical indicator of welfare for humpback and fin whales observed in a seriously anthropized feeding ground “. (Line 169- 172)

- “for photo-identification purposes and physical state characterization (Nikon D-800; 500/1000mm and 18/200m “. (Line 194-195)

-“ 2.4. Physical measures and welfare scoring system.  

Based on existing validated welfare assessment protocols in other species and in an intra-individual approach [7,32,39,55,129,130], we identified four categories of physical measures that could be assessed for both humpback and fin whales: body condition, skin condition, injury/scarring condition and parasite/epibiont condition. A scoring system was developed consisting of four different scales, one for each physical measure (three scales 0-4, and one scale 1-3) “. (Line 238-243)

- “Table 1. The physical indicator of welfare scoring system includes a 0-4 scale for body condition, skin condition, injury and scarring condition measures, and a 1-3 scale for parasite and epibiont “. (Line 272)

- “Although the majority of humpback whales were found to be in an overall good welfare state whilst fin whales were in a moderate one, the true potential of this first indicator to be included in a welfare assessment protocol will only be fully measured through long-term studies. Future research should focus on validating the current four measures of physical welfare, in other populations of humpbacks and fin whales “. (Line 585-589)

Comment 6, Introduction: Line 51: This is not the typical understanding in animal welfare science. I recommend that the authors read carefully Mellor 2016 and Mellor et al 2020 for an improved understanding of the subjective affective states referred to as welfare states.

Response 6 Introduction: Thank you for your comment. As Miller et.al. 2018 underlines, animal welfare scientists have a specific view of what the concept of Animal Welfare means and can be understood by the Three Orientation Models (see Fraser 2003, 2009 as referenced by Miller et.al. 2018) which is either a functional basis, a feeling basis (Like Marian Dawkin’s view) or a natural-lives basis (which is adopted mostly towards captive/domesticated animals). This “posture” is important to understand since it will influence the subsequent methodology used in assessing the welfare of the animals. We therefore clearly stated our view of animal welfare, which is a functional one, also used by multiple researchers and international associations like the American Veterinary Association or OIE. We subsequently suggest reading Mellor’s model (which is not a simple definition but rather a theoretical framework) at (line 89) to fully understand this specific approach based on the Five Domains and we also suggest the other model related to the Five Freedoms associated with the Welfare Quality protocols, which was used by Clegg et.al.2015 in the development of the C-Well protocol and is also the basis of our protocol.

Comment 7 Introduction: Lines 76-79: There are two key parts to a robust welfare assessment, the first is to validate a connection between the stressor and the impact, the second is the relationship between the impact and the mental state which is the animals welfare state.

Response 7: Thank you for your comment, we rephrased line 79-81 to take your comment into account and referenced Miller et.al.2018 (which is now reference 44):

“One of the key aspects of a scientific welfare assessment protocol is the holistic nature of its design which aims to establish relationships between animal-based indicators of welfare (physical, physiological and behavioural indicators, which include their respective measures) and environmental parameters that positively or negatively impact the affective state of the animal”. (Line 74-78)

Comment 8 and 9 Introduction:

Lines 80-83: Please delete the first sentence. The examples provided would not typically all be considered as health indicators e.g. body condition is usually used to infer impact on nutrition and lameness on behaviour. Lines 84: This is too simplistic a description of a welfare assessment. A single indicator of health is not robust enough to be able to provide a welfare assessment or score,

Response 8-9 Introduction: Thank you for your comment. We have rephrased the paragraph to better explain that an indicator (here the physical indicator is the general category as in Miller et.al. 2018 explain) includes selected measures like body condition or lameness, which will be assessed (by statistical validation through different tests of correlations, discrimination, etc.) to eventually infer welfare. The Body condition is a measure of the physical indicator/state of the animal, which includes the nutritional status but can also be related to the reproductive status or the sex of the animal (see line 100-107 with the references: 62, 64, 65, 68, 40, 42, 69, 70, 71, 72 and 73). You are right that lameness is often used as a measure included in a behavioural indicator of welfare, but it can also be included in a physical indicator since it shows physical/functional impairment like in the studies we have referenced: 48-49. Please read these references to understand. Here is the rephrased paragraph:

The first indicator to be validated in a welfare protocol is usually the physical status of an animal. Depending on the species and the context, the protocol may include a wide range of measures such as body condition score [43–45], lameness score [46,47], number and severity of injuries [48,49] and clinical observation of diseases [50,51]. Once the selected measures comprised in a physical indicator are validated statistically, they can be used to infer the underlying affective state through a scoring system [41]. On its own, the physical indicator can partially inform on the overall welfare of an animal but needs to be integrated into a complete species-specific welfare assessment protocol (see the Five Domains model by Mellor [41] or the Welfare Quality® protocols [31] (Line 79-87)

Comment 10 Introduction: Line 89-93: The start of this paragraph appears unrelated to the next section which seems more focussed and specific for the actual paper. Based on this I recommend the authors do further reading of recent literature on cetacean welfare where some studies have used some of the suggested indicators already in wild cetacean welfare assessments.

Lines 97-98: An indicator is one parameter that is measured i.e. you are suggesting 4 indicators of health, not a single indicator.

Line 171: Health and welfare are not separate, health is a facet of welfare.

Line 172: Physical health is not an indicator of welfare, it is a domain, each of the measures assessed in this study are indicators of health, which in turn can provide information to aid in the assessment of welfare.

Response 10 Introduction: Thank you for your comments which are extremely important. The use of the term’s indicator, parameters and measures seems to be misunderstood. We specified these concepts more clearly:  

One of the key aspects of a scientific welfare assessment protocol is the holistic nature of its design which aims to establish relationships between animal-based indicators of welfare (physical, physiological and behavioural indicators, which include their respective measures) and environmental parameters that positively or negatively impact the affective state of the animal” (Lines 74-78).

Comment 11 Methods: Lines 239-258: The instant decision to quantitatively score and weight various indicators without an understanding of the actual impacts on animal welfare is very concerning to me. Additionally, an arbitrary (used by the authors) percentage score to decide upon poor or good welfare lacks robustness and could be considered by those that do not consider animal welfare science highly to suggest that it does not have any backing. Please read Sandoe et al 2019 and carefully consider the concerns around aggregating multiple measures.

Response 11 Methods: Thank you for your comment. Welfare assessment protocols need to be validated statistically by measuring different animal-based measures. This is the foundation of an applied animal welfare assessment protocol. Our study is not a theoretical model/assessment of a situation but rather an applied welfare assessment protocol like the C-Well protocol developed for captive cetaceans (see Clegg.2015 and Botreau et.al 2007). Also, we did add Sandoe’s reference since he does point how intra-individual aggregation of measures is ethically much better (than inter-individual), which is exactly the approach we took by evaluating the four measures of physical welfare for each whale (50 fin whales and 50 humpback). We have added intra-individual in the paragraph:

Based on existing validated welfare assessment protocols in other species, and in an intra-individual approach [7,31,38,54,128,129], we identified four categories of physical measures that could be assessed for both humpback and fin whales” (Line 239-241)

Comment 12 Methods: Line 262: So this body condition scoring was not developed based on knowledge of the current species? indicator i.e.

Response 12 Methods: Thank you for your question. Yes, the body condition scoring was developed based on the knowledge of these species, however body condition scoring systems were non-existent in the literature, so we developed our scoring system based on extremely close species like the blue whale, whom is also a member of the rorqual family.

Comment 13-14 Methods:  Line 266: The dates appear to be arbitrary, you are trying to combine welfare status indicators with welfare alerting information into an aggregated measure. You would score two animals completely differently even if seen only a few hours apart, this does not appear to be scientifically robust and needs further deatil/clarification.

Line 269-272: Why do you feel that welfare is better in a lactating female in poor body condition? What is a female is not seen with a calf because it recently died? Or if a female is seen with a calf, but it is not her calf? A clear explanation must be provided as to why this change in score was given.

Response 13-14 Methods: Thank you for your comment. As you mentioned in your comment 8-9, body condition can be used to infer the nutritional status of the animal. In rorqual whales, they fast during the winter months and arrive in the St-Lawrence to feed in spring. An emaciated whale observed in June, might be related to the fact that this animal has arrived late in the feeding ground and might not have had the chance to feed properly yet. However, an emaciated whale observed in August would point towards a systemic health problem which has a greater negative impact on welfare than simply a lack of fat storage. In lactating females, the body condition is always thinner than non-lactating ones, but this is not due to poor welfare but rather because of the high energetic demand. If a calf dies, the female will stop lactating and her body condition will change rapidly. Rorqual whales are not gregarious, they would not be seen with the calf of another individual.

Comment 15 Methods: Table 1: You are providing a welfare score and inferring a welfare state along with potential affective states, but with no explanation of how these states were considered.

Response 15 Methods: Thank you for your comment. We explain how the different states were considered in sections: 2.4; 2.4.1; 2.4.2; 2.4.3 and 2.4.4.

Comment 16 Methods: Line 289-292: As stated in the introduction, you cannot observe the entire body of cetaceans in the water, how were % of body area assessed?

Response 16 Methods: Thank you for your question. We specify that it is the observed body surface, (Line: 291, 292, 294, 300, 302, 303, 326, 341)

Comment 17 Methods: Line 294: Is there information available on how these types of lesions impact upon cetaceans? These are only of welfare concern if there is an impact upon the animal that affects its subjective experience.

Response 17 Methods: Thank you for your question. Yes, we relate this information in the introduction (line: 122 to 169) and in the discussion, section 4.1 and 4.2.

Comment 18 Methods: Line 321: Are these welfare status indicators or welfare alerting indicators? How has this been considered when aggregating these measures?

Response 18 Methods: Thank you for your question. We have identified 4 animal-based measures (body condition, skin condition, injury condition and parasite/epibiont condition) that could be used in a physical indicator of welfare. Applied scientific animal welfare assessment protocols need to be evidenced based, not theoretically based like welfare alerting indicators.

Comment 19 Methods: Cookiecuttie scars/wounds: Why are these considered separately from the other types of wounds? What is the difference in the welfare impact being inferred from these different wound types? I would argue that the pain/discomfort that may be endured by the animal would not change based on where the wound came from (natural predator or anthropogenic) if the wound was otherwise identical.

Response 19 Methods. Thank you for your question. Cookiecutter wounds are part of the specific category of measure called: parasite/epibiont because the origin of the injuries needs to be differentiated to assess how anthropogenic activities impact welfare. We do recognize how cookiecutter wounds can cause pain and discomfort as stated from Lines 151 to 160: The relationship between the epibiotic organisms and their cetacean hosts can be one of parasitism or commensalism and can have different consequences on overall welfare of the whales [99,101,102]. When covering extensive parts of their hosts, diatom algae and barnacles like Coronula diadema are thought to impact swimming drag and hydrodynamics [100,103–105], but have a relatively small impact on overall welfare [101]. In comparison, cookiecutter shark bites (Insitus sp.), characterized by considerable epithelial removal, will leave open “cookiecutter shape” wounds that can range between 8 to 18 cm in diameter [106–108]. The welfare impact of such open wounds doesn’t only concern physical pain, but also susceptibility to infectious diseases and other sequelae [109–111].

Comment 20 Methods: Line 372: There is no information on how prey biomass were collected.

Response 20 Methods: Thank you for your comment.  At line 223, we give the website where all environmental dataset were retrieved with additional information from lines 234 -239, including reports in references 120, 121,123 and 125: We also retrieved annual biotic measures (data were collected in the Rimouski area, Anticosti, located in the Gulf of St-Lawrence, and Gaspé, on the south shore of the Gulf) on total zooplankton wet weight (g/m3) [121] and pelagic fish species abundance for sand lances (Ammodytes s.p), Atlantic herring (Clupea harengus), capelan (Mallotus villosus) and Atlantic mackerel (Scomber scombrus) which are the main dietary items of humpback and fin whales [120,123–125].

Comment 21 Results: Section 3.4: Please remove or re-word. You do not have the data and have not undertaken any robust inference to suggest welfare states. This could be re-worded to focus on the health impacts instead.

Response 21 Results. Thank you for your comment. We have rephrased: Aggregation of physical measures inferring overall welfare state   since this is specifically what we measured based on the description of our protocol in Section 2.4., 2.4.1, 2.4.2, 2.4.3, 2.4.4. (Line 467)

Comment 22 Discussion: Lines 482-491: Please re-write with the focus on your results being health indicators that could be used to help inform welfare assessments. I have provided a suggestion below: “Our study presents advances in validating a first physical welfare health assessment protocol for humpback and fin whales found in a highly anthropized environment, by analyzing over six thousand images and demonstrating positive correlations among four species-specific physical measures that can be used to partially assess health and welfare. Physical measures of welfare health in both species clustered in two categories, one related to physical state (body condition and injuries) and the other to the epidermal state (skin condition and parasites/epibionts).

Based only on the physical measures of welfare health, our results suggest that most humpback whales assessed in our study were in a positive welfare health state, compared to fin whales, who were in a moderate one. The main welfare health issues, in both species, were specifically related to two underlying factors, which could impact upon animal welfare: the cumulative effects of eutrophication on the environment, and direct anthropogenic activities associated with fishing activities and boat collisions. We discuss our results considering these two main potential welfare issues.”

Response 22 Discussion: Thank you for your comment and suggestions. Since we took your early advice to include the notion of health in the welfare concept (your comments: 1, 5 and 10) we have rewritten this section accordingly: 

Welfare assessment of wild cetaceans is increasingly gaining scientific attention as a preventive approach to inform international conservation bodies about priority concerns. For instance, in 2014, the International Whaling Commission (IWC) founded a working group on emerging welfare issues, resulting in the publication of a new theoretical framework for the welfare assessment of wild cetaceans and recommending further development of this framework based on field research [10,20]. Our study presents advances in validating a first physical indicator of welfare that could be included in an overall assessment protocol for humpback and fin whales found in a highly anthropized environment, by analyzing over six thousand images and demonstrating positive correlations among four species-specific physical measures that can be used to partially assess welfare. Physical measures in both species clustered in two categories, one related to physical state (body condition and injuries) and the other to the epidermal state (skin condition and parasites/epibionts).

Based only on the physical measures of welfare, our results suggest that most humpback whales assessed in our study were in a positive welfare state, compared to fin whales, who were in a moderate one. The main welfare issues, in both species, were specifically related to two underlying factors: the cumulative effects of eutrophication on the environment, and direct anthropogenic activities associated with fishing activities and boat collisions. We discuss our results considering these two potential welfare issues. (Line: 477-495)

Comment 23 Discussion: Line 505: Please provide a reference for this assumption since no histopathology was undertaken.

Response 23 Discussion: Thank you for your comment, we did provide references: 93-152 : “Skin diseases affecting cetaceans can be caused by several pathogens including viruses, bacteria or fungi, and have been correlated with environmental parameters such as low salinity levels and high water temperatures [93,152]. “(Line 506-508)

Comment 24 Discussion: Line 509: Please change ‘welfare’ to health.

Response 24 Discussion: Thank you for outlining this but we cannot replace the word welfare for health in this sentence since we explicitly mention health in the overall welfare “the importance of skin health on overall welfare” (Line 519-520)

Comment 25 Discussion: Line 525: Please re-word to read ‘could impact negatively on cetaceans’ health leading to poorer welfare

Response 25 Discussion: Thank you for your comment. In accordance with your previous comments that health is an integrated part of welfare (notably comment 1, 5, 10 and 22) we have removed health from our manuscript to focus only on the general concept of welfare which you pointed out, includes health. This is the re-wording of this section: “Furthermore, higher sea temperatures seems to increase the survival of some pathogens that could impact negatively cetaceans’ welfare but more studies on this potential problematic are needed [158,159].” (Line 525-526).

Comment 26 Discussion: Line 535: This is an example of a result where the authors could highlight the impact on welfare. Therefore, I suggest that they delete ‘health and’

Response 26 Discussion: Thank you, again this is an excellent comment highlighting that health should always be included in the concept of welfare as you mentioned previously. We have modified accordingly: “This finding highlights the long-term effects of severe anthropogenic injuries and, besides direct physical impairment due to injuries, may suggest a chronic stress response impacting their overall welfare”. (Line 536)

Comment 27 Discussion: Line 549-550: This is another example where the authors could point out that this is an impact on welfare, which would better make the point of the inextricable link between conservation (survival) and welfare. Lines 555-557: As above, this is an example where the authors could be pointing to negative welfare impacts.

Response 27 Discussion: Thank you for your comment. By including these examples in our discussion, we feel that the readers will automatically make the link between conservation and welfare, as it is implied.

Comment 28 Discussion: Line 558: Health is a facet of welfare, please remove the ‘health and’.

Response 28 Discussion: Thank you again for pointing out this inconsistencies, we have edited the paragraph: “This negatively affects fitness and the immune response through the continued release of glucocorticoids and increases the risk of infectious diseases [166,167]. Our study further confirms that acute and chronic entanglements in fishing gears represent a serious threat to the welfare of North Atlantic whales”. (Line 559).

Comment 29 Discussion: Lines 561-562: This statement is not correct, there are recent publications which have undertaken welfare assessments in free ranging cetaceans. Additionally, this manuscript has not produced a welfare assessment protocol. This statement must be removed.

Response 29 Discussion: Thank you for your comment, we do agree with you and have changed the sentence to: “This study is the first to validate a physical indicator of welfare in free-range mysticetes like fins and humpback whales by correlating four animal-based physical measures. (Line: 561-562).

Comment 30 Discussion: Line 562-563: It is not clear how the method used is reliable or valid, further detail is needed to clarify this statement.

Response 30 Discussion: Thank you for your question, we have detailed our statistical methodology in length in the section 2.4.(lines: 241-260) and section 2.4.5 (Statistical Analysis (line 351-388).

Comment 31 Discussion: Line 567: Samples of skin were not taken; therefore you cannot determine definitively that skin disease was present. Please alter the wording accordingly.

Response 31 Discussion: Thank you for your observation, however, here we simply underline our statistical results and the significant relationship between our measures, which one of these is the umbrella category of skin diseases.

Comment 32 Strength and limitations: Section 4.3: This entire section should be re-written, there is no need to be repeating results that have just been considered in the discussion. Instead, a focus on the overall achievement of the paper i.e. ability to assess health indicators related to skin/external body from visual assessments should be made. The limitations of the study i.e. not being able to definitively assess aetiology of skin lesions. Then highlighting how this could help in the development of a welfare assessment. But please take careful consideration in the explanation of how a welfare assessment could be developed for these species by undertaking signicantly more reading of the animal welfare science literature.

Response 32 Strength and limitations: Thank you for your comments and suggestions. We do not entirely agree with your suggestions about this section since we believe it is important to recognize the limits of our study, even if we repeat certain aspects of our results and discussion. This section is not mandatory in the journal, but we feel it is important to show that we are able to recognize our strengths and mostly, our limitations, showing strong ethical and scientific professionalism.

Regarding your comment on developing further our literature review on the different concepts and theories on animal welfare science, we can understand your point of view since our manuscript did not develop the theoretical foundations underlying our welfare assessment protocol. This is why we have suggested some further readings (Lines 84-87), so the readers can look up at least 2 different theoretical approaches, which are prior knowledge to understand an applied welfare assessment protocol.

We strongly agree with the general comments you made in this review, mentioning:” Animal welfare science is a new, emerging area in marine mammal science which has often had many diculties in being accepted, due to many scientists misunderstanding the ‘ease’ of assessing and inferring welfare states. We couldn’t agree more with you on this statement, and this is why, we needed to validate scientifically (statistically) different measures of physical state (which we defined as the umbrella category: physical indicator). As you mentioned, the misunderstanding about welfare assessment and thinking it’s an ‘easy’kind of assessment is not helping in the recognition of animal welfare as a scientific discipline and that’s why we have developed this first (part) of a protocol and stretched to validate it with statistical analysis.

Comment 33: Line 579: Please change this sentence to be related to the actual aim and results of the paper i.e. assessing physical health indicators and how these potentially relate to environmental parameters.

Response 33: Thank you for your comment although this sentence is not about the aims of our study but rather the limits of our study:

“Finally, our physical indicator protocol can only partially assess welfare since it needs to correlate with behavioural and physiological indicators to validate the overall welfare status of an animal. The inevitable subjective aspect of our measuring scales’ thresholds (negative/moderate/good) can only be objectively validated over time when correlated with other measures.”

Comment 34:

Conclusion: Please re-write the conclusion with careful consideration of all the points raised above, the paper has assessed four indicators of health, this alone cannot provide a welfare assessment.

Response 34: Thank you for your comment. We have indeed assessed four animal-based measures that statistically are valid because they positively correlate but also measure discriminately four aspects of the physical state of the animals. Our conclusion simply reflects the statistical results, which we cannot change, they are not opinions, they are statistical facts. However, we have reformulated/rewrote the conclusion to take your comments into account:

5. Conclusion

The purpose of this study was to statistically validate a first indicator of welfare for humpback and fin whales living in an anthropized environment. Based on a multi-scale scoring system of body, skin, injury, and parasite/epibiont condition measures, our results showed positive inter-correlation and discrimination between all measures validating our physical indicator. Overall welfare states, for both humpback and fin whales, were mostly impacted by the degradation of the marine environment and previous physical trauma due to anthropogenic activities. Although the majority of humpback whales were found to be in an overall good welfare state whilst fin whales were in a moderate one, the true potential of this first indicator to be included in a welfare assessment protocol will only be fully measured through long-term studies. Future research should focus on validating the current four measures of physical welfare, in other populations of humpbacks and fin whales, but also in other species like the gray whales, who are showing signs of chronic stress and poor health. Finally, the measures included in our physical indicator of welfare need to be correlated with other measures/indicators like physiological measures of stress and behavioural responses. (Line 578-593)

Comment 35 References: Key important references that have undertaken cetacean welfare assessments and/or considered how to undertake welfare assessment in mammals are missing in the paper: Please undertake further reading e.g.,

Response 35 References: Thank you for suggesting further references, we will briefly comment on each one as regards to their relevance for our study.

  • Boys et al. 2022: https://royalsocietypublishing.org/doi/full/10.1098/rsos.220646

This study, as the title explicitly mentions, aimed to identify potential welfare and survival indicators through expert opinions. This study didn’t aim to validate specific measures with scientific statistical analysis, but rather to base its conclusions on expert opinions, so not relevant to our study.

  • Boys et al. 2023: https://onlinelibrary.wiley.com/doi/full/10.1111/mms.13029

This study is interesting and based explicitly on Mellor’s theoretical model. However, here, the animals are stranded so their welfare is clearly compromised, and they can observe the whole body of the animals and their behaviour second by second compared to free-ranging cetaceans. It is irrelevant to our study since we didn’t choose Melor’s model to develop our protocol but rather the Welfare Quality approach used by Clegg. 2015.

  • Serres et al. 2024: https://pubmed.ncbi.nlm.nih.gov/36317250/

This is an interesting paper on behavioural measures of welfare on captive cetaceans. It is not relevant to free-range species and physical measures of welfare.

  • Miller et al. 2018: This is an excellent paper and quite relevant to our study and we have integrated this reference: https://www.aquaticmammalsjournal.org/article/vol-44-iss-2-miller/

  • King et al.2021: Interesting paper on whales observed elsewhere. There is no welfare assessment and so irrelevant to our study : https://www.tandfonline.com/doi/abs/10.1080/17451000.2021.1967993

  • Clegg et Al. 2017: We already referenced Clegg (2015) and this article (2017) is interesting but is a review aimed at developing bottlenose welfare assessment protocol, which is a member of the odontocete sub-order of cetaceans, with different biology and behaviours than rorqual whales. Not irrelevant but adding this reference would not improve our manuscript : https://www.researchgate.net/publication/316855744_Applying_welfare_science_to_bottlenose_dolphins_Tursiops_truncatus

  • Beaulieu 2023: impossible to find this reference.

  • Sandøe et al. 2019: This is an excellent paper and Peter was my professor, his critical thinking is important in the field of animal welfare science. This paper is partially relevant as it is specifically directed at commercial farm settings. He does point how intra-individual aggregation is ethically much better (than inter-individual), which is exactly the approach we took by measuring the four measures of physical welfare for each whale (50 fin whales and 50 humpback). We have therefore added the reference 130: (Line 240) https://www.cambridge.org/core/journals/animal-welfare/article/aggregating-animal-welfare-indicators-can-it-be-done-in-a-transparent-and-ethically-robust-way/EAD62DD23305804EC1586497292955E6

  • McMahon et al. 2012: we could not find this paper.
  • Several chapters in Butterworth 2017 e.g: We have referenced Butterworth on 3 occasion in our manuscript. We did not reference the chapters of this book, as they were more general concepts.

  • Fernandez et al. 2017: this is a paper on the distribution of cetaceans in a different region of the world irrelevant to our study: https://www.frontiersin.org/journals/marine-science/articles/10.3389/fmars.2021.688248/full

  • Harley et al. 2021: the only paper found is from 2022 and irrelevant to our study : https://link.springer.com/article/10.1007/s10071-022-01679-5

  • Beausoleil et.al.2018: We have referenced Beausoleil on four occasions et our manuscript.

  • Lesimple 2020. Interesting review but not specifically relevant to our manuscript: https://www.mdpi.com/2076-2615/10/2/294

  • Cassoff et.al.2011. Interesting article, but we have referenced multiple articles on entanglement which were specifically relevant to the long-term effects on welfare.https://www.int-res.com/abstracts/dao/v96/n3/p175-185/

  • Dolman and Brakes 2018. This is an great review, but we do not feel it would enhance our manuscript. https://www.frontiersin.org/journals/veterinary-science/articles/10.3389/fvets.2018.00287/full

  • 2013. We have referenced Broom already in our manuscript.

  • Segura-Göthlin et al. 2021. Interesting methodology but since we didn’t use this in our study, it is irrelevant to reference it. https://www.ncbi.nlm.nih.gov/pmc/articles/PMC8532937/

  • Browman and Skiftesvik 2011. This is an extremely important opinion-based communication on best practices in science. We agree with the authors and always take these possible biases into account:https://www.int-res.com/abstracts/dao/v94/n3/p255-257

  • Papastavrou et al. 2017. This is an excellent paper and we believed we had included this one in our introduction. We included this reference 11: (Line 44) https://www.sciencedirect.com/science/article/abs/pii/S0308597X16306960

Round 2

Reviewer 2 Report

Comments and Suggestions for Authors

Comments on the Quality of English Language

Minor editing for grammar and spelling required

Author Response

Reviewer General comment 1:

The authors have made some good changes to the manuscript, there are however several parts that still require addressing. I have replied/commented on these in reply to the authors below.

Notably, the position on welfare that the authors wish to take and from which they have developed this manuscript needs to be stated, currently this appears to be a mixture which confuses the messaging around welfare. My other concern is that the authors seem to have dismissed the idea of undertaking further reading and consideration of additional relevant literature in their manuscript. I believe the authors should undertake detailed, careful reading of the suggested literature to understand the relevance of their work to the wider marine mammal and welfare literature. It is still not clear how weighting of the different indicators were scored and aggregated and the limitations of this approach have not been discussed, despite the importance of considering all indicators as equally impacting upon welfare, as highlighted by Sandoe et al. These points must be addressed before the manuscript is published.

Response General comment 1:

Thank you again for reviewing a second time our manuscript. You are absolutely right concerning the fact that our manuscript does not clearly state one theoretical framework underlying our applied assessment protocol, since the aim of our study was not to demonstrate how theoretically we should develop a welfare assessment, but rather to develop an actual protocol based on multiple theoretical references, which implicitly define our epistemological foundations. However, we have reviewed our introduction to state this more explicitly: “In this regard, our epistemological posture is based on theoretical frameworks developed for structuring coherent assessment protocols (the Five Freedoms Model [33,34] and mostly the Five Domains Model [35–37], which have been applied to farm [38,39], companion [40,41], captive wildlife [42,43], laboratory animals, and more recently to free-ranging species impacted by anthropogenic activities [44–47]. Common welfare assessment principles include the necessity to engage in a holistic approach to identify relationships between animal-based indicators of welfare (physical, physiological and behavioural indicators, which include their respective measures) and environmental parameters that positively or negatively impact the affective state of each animal [21,35,37,48–50]”. (Line 68-77)

Concerning the weighting of the different measures, not the indicators since here we have different measures included in one physical indicator (this is based on our terminological structure where we have umbrella animal-based indicators: physical, physiological and behavioural indicators, which include species-specific measures: body, skin, injury and parasite/epibiont categories) we have detailed extensively this aspect in the methodology notably by referencing multiple methodological papers, including Peter Sandoe, and by providing detailed photographs and explanations concerning the four different scoring scales. We apologize if we gave the impression, we didn’t carefully review the literature you proposed. Here are some reflections we have about Sandoe (2019) and Mellor (2009, 2015 and 2020) on the aggregation and scoring of our four measures:

Sandoe (2019) starts his article by explicitly mentioning “A central aim of animal welfare science is to be able to compare the effects of different ways of keeping, managing or treating animals based on welfare indicators. A system to aggregate the different indicators is therefore needed.” Furthermore, Mellor’s (2020) summary of the grading methodology (section 3.2) states: “A five-tier scale (A to E) is used to grade negative welfare impacts according to the presence, intensity and/or duration of specific negative affects. Thus, grades A and B represent no and tolerably low-level impacts respectively, grade E represents very severe negative impacts related to experienced affects variously manifesting at high to very-high intensities and/or for long to very-long durations and grades C and D represent intermediate-level impacts related to their intensities and/or durations. These grades therefore equate to different degrees of welfare compromise, ranging from none to very severe” Mellor (2020) explicitly mentions that the Five Domains Model is not intended to define good and bad welfare, nor is it intended to accurately depict body structure and function. Rather, it is a device for facilitating systematic, structured, thorough and coherent assessments of animal welfare, and for qualitatively grading welfare compromise and enhancement (see Section 3.3) [6,7,16]. The purpose of each domain is to draw attention to areas that are relevant to welfare assessments”.

In this regard, we have used a numerical grading system simply because it is more convenient for some statistical tests, but it is still a qualitative based scoring system. For instance, if we look at our body condition measure scoring scale (section 2.4.1) we have different levels of severity, graded for instance in the case of emaciated animals with 0-1, which represent level E for Mellor: “represents very severe negative impacts related to experienced affects variously manifesting at high to very-high intensities and/or for long to very-long durations”. We developed our grading system based on species-specific knowledge and the inherent structure of our protocol only reflects that fact. We did recognize that our numerical grading system was arbitrarily based on the worst and best scores (Line 256-257) and, we did mention in the limits of our study that the subjective nature of our scoring system can only be objectively validated over time: “Finally, our physical indicator can only partially assess welfare since it needs to correlate with behavioural and physiological indicators to validate the overall welfare status of an animal. The inevitable subjective aspect of our measuring scales’ thresholds (negative/moderate/good) can only be objectively validated over time when correlated with other measures”. (line 575-579).

When going deeper in Sandoe’s ethical reflection (mostly related to round 1 comments) about aggregation of measures, we absolutely agree with him on how different ethical dilemmas can arise in the context of farm animals or other animals under our care. However, here, we are assessing the welfare of free-range cetaceans, we are not in an utilitarian ethical approach nor any specific one since our goal is not to weight the welfare of the whales in comparison to the welfare of humans to take decisions. We have no ethical dilemma since we are not “using” the whales. The only ethical consideration is one about the “duty of care” towards wild sentient animals so stakeholders can prioritize mitigation measures. In this context, the aggregation of measures (measuring scales, indicators, parameters, etc.) do need to be aimed towards assessing the welfare of individual whales, so we can subsequently inform on the % of animals having poor welfare due to anthropogenic activities. Animal welfare science needs to be as objective as possible (even if we are partially in a qualitative methodology and that there are subjective dimensions to the scoring of welfare states) to simply inform stakeholders. For instance, we absolutely agree with Sandoe when he references Veissier et al. (2011) : “According to some influential ethical theories[…],the relative negative load on the welfare of individual animals, and not just the sum of welfare scores across individuals, matters ethically. Therefore, it is important to aggregate welfare at the level of the individual animal. However, the short answer to this question is that it does not happen in the WQ protocols. This is because WQ aggregates across indicator values without considering the welfare status of individual animals” (Veissier et al 2011). This is exactly the approach we took (to assess at the individual level) when developing this first Physical indicator assessment protocol.

Finally, we have looked a second time at the literature you proposed, and we have included three articles from Boys et.al (ref.46-47-48, line 788-794) and one from Serres (ref. 20, line 726), as they are important studies, and we agree they do need to be referenced.

Reviewer Comment 2 Title:

Please delete the word ‘first’ in the title

Response 2 Title:

Thank you for your comment, we have changed the title (Line 4)

Reviewer Comment 3 Abstract:

Apologies if this was not clear previously, I meant to state whether the example you gave for instance was related to higher prevalence of lesions when there was lower dissolved oxygen?

Response 3, Abstract:

Thank you for rephrasing, we didn’t want to specify this in the abstract since higher prevalence of pale skin patch was associated with higher dissolved oxygen levels, but lower dissolved oxygen was associated with higher Tattoo skin disease-Like in fin whales, and so on, etc.

Reviewer Comment 4: Introduction

 Yes, this is absolutely correct, however, in the current manuscript you have not stated what “posture” you are taking and seem to jump between one which is functional only and one which is feeling based. In this way you are following slightly more along the lines of the Five Domains Model which considers that cumulative impacts on the functional domains have an impact on affective state which is what provides an understanding of welfare state. Neither is necessarily right or wrong, but please re-consider how you have written and discussed the concept of welfare throughout the manuscript and provide an explicit explanation of what “posture” you are taking.

Response 4: Introduction

Thank you for your comment. We have defined our posture by using two different definitions which, as you mentioned, have both functional and feeling based characteristics, which aligns with current animal welfare conceptions. We have further developed our “posture” by detailing frameworks and basic welfare principles (Line 68-77).

Reviewer Comment 5: Methods

 I think you misunderstand what is meant by welfare alerting indicators, these are not theoretical indicators, but those that present a potential risk or opportunity for welfare.

Response 5: Methods

Apologies for misunderstanding your question. The concepts associated with welfare alerting indicators can vary from one author to the other. For instance, Delfour et al. (2020 https://www.mdpi.com/2673-5636/1/1/4) has identified six alerting factors associated with a behavioural state called Willingness to participate. You are clearly referring to Boys et al. (2022): “ Only animal-based indicators can provide direct information about the animal’s ‘welfare status’ and are often preferred in welfare assessments. However, some animal-based indicators may only be ‘welfare alerting’ in that they can indicate a predisposition for welfare impacts that relate to the animal itself rather than its environment, for example, an animal that is neonatal or unweaned. Welfare alerting indicators are generally more feasible and reliable to assess across time and different observers and are often non-invasive. They are therefore commonly applied in welfare assessments.” Based on this conception, we only included one animal-based characteristic relating to a welfare alerting characteristic, which was lactating females. As we mentioned in the methods section, we only used the photographs of each animal observed for the first time.

Here then is a revised response to your earlier comment about body condition scoring: “The dates appear to be arbitrary, you are trying to combine welfare status indicators with welfare alerting information into an aggregated measure. You would score two animals completely dierently even if seen only a few hours apart, this does not appear to be scientically robust and needs further deatil/clarication.” In the case of body condition scoring, we are taking into account the lactating status of the female, which *could* be seen as a welfare alerting indicator (if e.g. we consider that lactating females are more at risk than non-lactating females). But we are not actually using lactating/non-lactating status as a welfare indicator (neither alerting nor status). We do not infer that the whale’s welfare is worse because she is lactating. We are simply using that information to set a sensible baseline for what is considered a healthy body condition. We assume that welfare is compromised if a whale is skinnier than what would be normal given her physiological status (lactating or not) and given the time of year, to account for normal fluctuations in body condition based on lactating status and season. It is common in assessments of farm animal welfare to acknowledge that normal or ideal body condition varies in this way, see e.g. for cows https://www.ontario.ca/page/body-condition-scoring-dairy-cattle, which states that “ideal condition scores are 3.0–3.25 at dry off and calving, and 2.25–2.75 at peak lactation”.

You are entirely correct that the cutoff date chosen is arbitrary and may result in different scores between similar animals observed hours apart; this arbitrary cutoff issue however applies to every qualitative level-based grading scale (e.g. in the cow body condition scale linked above, scores vary from 1 to 5 and the boundary between any two adjacent levels was chosen according to sensible yet ultimately arbitrary criteria, which could have been chosen slightly differently).

Reviewer Comment 6: References

.

Whilst I understand that the authors may not feel that inclusion of all the previously suggested references were necessary, I would caution their response that the published literature were “not relevant to our study”. Specifically, this study has stated that the welfare model underlying it is the C-Well model which was developed for captive bottlenose dolphins. Using the authors own argument, that means the model used is also not relevant for their study as they have researched free-ranging rorquals. Additionally, in the C-Well model much more of the body area was observable on the dolphins since they were captive. Notably, in this sense I question why the authors have decided to focus upon applying a model developed for captive cetaceans rather than the model developed for free ranging wild cetaceans by Nicol et al? In the literature that I previously suggested and will re-iterate here there are multiple reasons for relevance to the manuscript: Boys et al. validated and assessed welfare indicators, some of which are similar to those used in this manuscript (body condition, injuries). King et al 2021 developed a welfare assessment tool for NARW. Segura- Göthlin et al. specifically examined skin conditions in cetaceans comparing between visual observations (method you have used) and the actual aetiology of the lesions via histology. Harvey et al. developed a method for assessing indicators of welfare in free ranging animals. I strongly argue that there are important points in all of these published papers which would further support/strengthen the work being undertaken. The suggestion by the authors that they are just not relevant because they are not the same as their study is disappointing.

Response 6: References

Thank you for your comment and pointing out your disappointment. We apologize if we sounded that we disregarded the literature you suggested and as we explained in response 1, we have reviewed the literature you proposed initially and have included some of these references in our latest manuscript version. We will comment here a second time on the proposed literature, with greater attention. However, we want to mention that more than a few references we found might be the wrong ones, since you didn’t reference them in a conventional way, but rather by simply stating the name of the first author and the year. Some authors have published more than one article in the same year, and we feel we were not able to fully appreciate the literature review you proposed. We therefore might have replied briefly since our research didn’t retrieve the right source you were referring to. Here are our new comments considering the results of our research based on the basic information you gave us:  

  • Boys et al. 2022: https://royalsocietypublishing.org/doi/full/10.1098/rsos.220646

This study is indeed important, especially because it shows a first step in validating potential survival and welfare indicators through expert opinion, using a Delphi approach/method. We feel that this is a “gold standard” especially for researchers who have little experiences with either species-specific knowledge or when applying welfare assessment as a new methodology. We have included this reference and Boys https://www.mdpi.com/2076-2615/12/14/1861  (Ref 46-47, Line 788- 793)

  • Boys et al. 2023: https://onlinelibrary.wiley.com/doi/full/10.1111/mms.13029

This study is interesting and based explicitly on Mellor’s theoretical model. We have included this reference since it details the use of a theoretical framework, which shows the benefits of using such model in the development of welfare assessment protocols in free-range cetaceans, here, specifically for stranded groups of cetaceans. Although not directly associated with our own methodology, this study is extremely important for the wider development of welfare assessment protocols, especially regarding the almost “non-existent methods” currently available. We feel that Boys should pursue this line of study so it can be implemented on an international scale. Here in Quebec, we’ve had many problems due to the lack of understanding of welfare assessment of stranded whales. We have referenced this article (Ref 48, Line 794)

  • Serres et al. 2024: https://pubmed.ncbi.nlm.nih.gov/36317250/

(Our original response: This is an interesting paper on behavioural measures of welfare on captive cetaceans.

Initially it seems we didn’t have the right article you were referencing. After further researching, we think you were referring to this article: https://www.researchgate.net/profile/Songhai-Li/publication/378401574_Selection_of_parameters_to_assess_the_welfare_of_free-ranging_Indo-Pacific_humpback_dolphins_using_expert_opinion_survey/links/65dd2ad2c3b52a1170fbcec0/Selection-of-parameters-to-assess-the-welfare-of-free-ranging-Indo-Pacific-humpback-dolphins-using-expert-opinion-survey.pdf

We did include the reference (the one we shared) in our manuscript (Ref 20, Line 726)

  • Miller et al. 2018: This is an excellent paper and quite relevant for our study and we have integrated this reference: https://www.aquaticmammalsjournal.org/article/vol-44-iss-2-miller/

Integrated (Ref 51 and we had already 83, Line 800 and 890)

  • King et al.2021: Interesting paper on whales observed elsewhere. There is no welfare assessment, maybe this is the wrong reference, it is difficult to see how this article could contribute further our study : https://www.tandfonline.com/doi/abs/10.1080/17451000.2021.1967993

  • Clegg et Al. 2017: We already referenced Clegg (2015) and this article (2017) is interesting but is a review aimed at developing bottlenose welfare assessment protocol, which is a member of the odontocete sub-order of cetaceans, with different biology and behaviours than rorqual whales. Not irrelevant but adding this reference would not improve our manuscript : https://www.researchgate.net/publication/316855744_Applying_welfare_science_to_bottlenose_dolphins_Tursiops_truncatus

  • Beaulieu 2023: impossible to find this reference.

  • Sandøe et al. 2019: This is an excellent paper We have added the reference: https://www.cambridge.org/core/journals/animal-welfare/article/aggregating-animal-welfare-indicators-can-it-be-done-in-a-transparent-and-ethically-robust-way/EAD62DD23305804EC1586497292955E6

(Ref: 134, Line 1020)

  • McMahon et al. 2012: we could not find this paper.

  • Several chapters in Butterworth 2017 e.g: We have referenced Butterworth on 3 occasions in our manuscript. We did not reference the chapters of this book, as they were more general concepts.

  • Fernandez et al. 2017: this is a paper on the distribution of cetaceans in a different region of the world. This might not be the one you were referring to: (https://www.frontiersin.org/journals/marine-science/articles/10.3389/fmars.2021.688248/full

  • Harley et al. 2021: the only paper found is from 2022. This might not be the one you were referring to : https://link.springer.com/article/10.1007/s10071-022-01679-5

  • Beausoleil et.al.2018: We have referenced Beausoleil on four occasions in our manuscript.

  • Lesimple 2020. Interesting review but not specifically relevant for our manuscript since it is focused on horses. We have specifically referenced Harvey (who worked on wild horses) but of course we could add this one. Maybe you could explain what you feel this paper would add beyond Harvey’s? https://www.mdpi.com/2076-2615/10/2/294

  • Cassoff et.al.2011. Interesting article, but we have referenced multiple articles on entanglement which were specifically relevant to the long-term effects on welfare. We are unsure what this reference would contribute to our study beyond the ones already cited: https://www.int-res.com/abstracts/dao/v96/n3/p175-185/

  • Dolman and Brakes 2018. This is a great review on problematics of bycatch/entanglements in cetaceans, but we do not feel it would enhance our manuscript (only add to an already HUGE reference section, since we already have 177 references, including some on the problematic of entanglement/bycatch: https://www.frontiersin.org/journals/veterinary-science/articles/10.3389/fvets.2018.00287/full

  • 2013. We have referenced Broom (multiple articles/editions) already in our manuscript.

  • Segura-Göthlin et al. 2021: Segura-Göthlin et al. 2023 is in our reference list. We consider it more appropriate for our study, as they studied several types of skin lesions in cetaceans. We cite other papers reporting on the validation of visual diagnosis through histological and molecular studies (Murdoch et al. 2008, Blacklaws et al. 2013, Duignan et al. 2020).

  • Browman and Skiftesvik 2011. This is an extremely important opinion-based communication on best practices in science. We agree with the authors and always take these possible biases into account: https://www.int-res.com/abstracts/dao/v94/n3/p255-257

  • Papastavrou et al. 2017. This is an excellent paper and we believed we had included this one in our introduction. We included this reference: https://www.sciencedirect.com/science/article/abs/pii/S0308597X16306960

(Ref 11 (Line 709)

Round 3

Reviewer 2 Report

Comments and Suggestions for Authors

See Word document attached

Comments on the Quality of English Language

Minor changes needed

Author Response

Revier Comment 1

Thank you for further defining your position when considering how you are undertaking this welfare assessment. This is helpful for readers to better understand the framing of your work and results.

While I am happy with how this is now addressed in the paper, I would recommend that the authors spend some time thinking through their ideas and position on welfare assessments. In the first review, the authors appeared quite adamant that they were not following the Five Freedoms or Five Domains in their approach to welfare assessment. However, in this review iteration they have now stated that this is exactly what they have based the assessment on. There is no single right way to assess welfare, but it is important that all thinking is transparently explained to ensure that the results and implications of a study can be fully understood and contextualised.

While I appreciate the explanation around the scoring applied to each measure, I still do not feel that the authors have fully understood and adequately accounted for what this means in the study. I understand that as part of the aim of the study it has been important to statistically evaluate the indicators alongside various parameters, however due to the way in which these measures have been scored and aggregated the authors have suggested that all welfare measures are equal and will have equal bearing on welfare. This is possible but is not likely to be biologically true. I am not suggesting that the authors should not aggregate measures, but they do need to carefully consider what it means for their assessment if they are stating that all factors have equal bearing on measure and some thought and discussion around this is necessary.

Response 1:

Thank you for your comments and suggestions.

The measures included in our scoring scheme have all different weights, as detailed in section 2.4, table 1. So, we are not sure exactly what you are referring to when you say “due to the way in which these measures have been scored and aggregated the authors have suggested that all welfare measures are equal and will have equal bearing on welfare

First, our scoring system was developed as to assess each measure based on a severity levels approach (instead of a dichotomous approach of absence/presence), detailed in Table 1: The physical indicator of welfare scoring system includes a 0-4 scale for body condition, skin condition, injury and scarring condition measures, and a 1-3 scale for parasite and epibiont condition scores. The score of each measure was then transformed to a percentage scale and weighted according to the CRITIC method. The final score for each whale was then associated with the corresponding inferred welfare state. (Lines 273-277)

We used the Criteria Importance Through Inter-Criteria Correlation method (CRITIC) to determine the weight of each measure [135,136]. This method is used to combine multiple criteria into one overall score by attributing a larger weight to the measures that have a greater standard deviation, and that are negatively correlated to other pairs of criteria. (Line 245-249).

If you are referring to section 3.4, Aggregation of physical measures inferring welfare and Table 5, these are the results after we applied our Critic approach as we mentioned in section 2.4: “We used the CRITIC method separately for fin and humpback whale measurement scales, as the results showed different weights should be used for each criterion in each species (Table 1). The final percentage grading scale was used to infer (arbitrarily based on the worst and best scores) the overall welfare state of each animal, as follows: a score between 1 and 40% was considered to represent poor welfare, a score > 40% up to 75% was considered moderate welfare, and a score > 75% was considered good welfare (see Table 1). (Line 253-258).

Reviewer Comment 2

I would also suggest while the authors may not be focusing on a specific ethical position, the underlying purpose based on their introduction, discussion and replies to reviewer comments is in fact one of a utilitarian approach. However, I was not suggesting that an ethical position needed to be stated, but that an understanding and discussion of how their scoring and aggregation method may introduce bias, as explained above.

Response 2

Thank you for your comment. We are fully aware of biases and that’s the reason we chose the CRITIC method to weight our different measures.

Reviewer Comment 3

I am pleased that the authors have finally taken the time to better look at the literature and have found some additional studies of relevance to their work. I would like to caution the authors in the future from immediately assuming that literature is not relevant just because the context was different.

Author point: Thank you for rephrasing, we didn’t want to specify this in the abstract since higher prevalence of pale skin patch was associated with higher dissolved oxygen levels, but lower dissolved oxygen was associated with higher Tattoo skin disease-Like in fin whales, and so on, etc.

In this case then, do you really think that there is a biologically meaningful impact of dissolved oxygen on your animals? How can you then pull these measures together when considering skin condition?

Response 3

Thank you for your comment.

You will see that there are some statistical significances between oxygen levels and different cutaneous lesions reported in our results: “abiotic environmental parameters did have significant relationships with some skin diseases and parasite/epibionts in both fins and humpback whales (Table 4). Low saturated oxygen levels (DO) (M = 17.06, SD =17.93), mostly in the Rimouski area, were correlated with high levels of nitrate (r = -0.49, p ˂ .001), phosphate (r =-0.74, p ˂ .001), chlorophyll A (r = -0.39, p =.006), sea temperatures (r = -0.52, p ˂ .001) and salinity levels (r = -0.56, p ˂ .001) consistent with eutrophication. A total of six cutaneous lesion categories were related to environmental parameters in one of the two species. Light focal skin disease, tortuous cutaneous disease, and cookiecutter wounds were not correlated to any environmental parameters (for both species). Finally, one category (pale skin patch syndrome) was related to higher dissolved oxygen levels in both species” detailed in Table 4, (Line 453-461).

We then discuss these findings in section 4.1: “Skin diseases affecting cetaceans can be caused by several pathogens including viruses, bacteria or fungi, and have been correlated with environmental parameters such as low salinity levels and high water temperatures [98,157]. However, these two measures are also intrinsically correlated to other marine environmental parameters, notably: dissolved oxygen levels, integrated nitrate, phosphate and chlorophyll-A levels. Our study showed that, between 2016 and 2021, dissolved oxygen levels in the St-Lawrence estuary reflected hypoxia and were highly correlated with all other environmental parameters suggesting eutrophication [158–160]. We found that pale skin patch syndrome prevalence was observed in both species when dissolved oxygen levels were higher, suggesting a non-pathological epidermal condition like desquamation. Epidermal growth and renewal is rapid and continuous in cetaceans and actively maintains a protective barrier from environmental stressors [161]. Furthermore, epidermal homeostasis disruption has been linked to systemic imbalance of nutrient levels, like iron, in aquatic and terrestrial animals, underlying the importance of skin health on overall welfare [162,163]. Conversely, lower dissolved oxygen levels were correlated to the prevalence of tattoo skin disease-like lesions in fin whales, suggesting that this environmental parameter might have greater impact on cetacean skin health than previously thought. This hypothesis should be further explored.”  (Line 503-520).

Reviewer Comment 4

Now that you have realised what was being referred to as alerting indicators (based on Harvey et al. papers), how do you think such alerting indicators can be used in combination with those that are welfare status indicators? Should they be scored/weighted in an equal manner and if this is what you choose to do (which is as current) what are the implications? It is very important that an understanding of what this means for your results are fully understood and transparently explained.

 Response 4

Thank you for your comment.

As mentioned previously, our study is focused on the application of a welfare assessment and therefore, we have assessed only animal-based measures because we are interested in the animal’s welfare state as it is when we observe the animal. However, welfare alerting indices, as defined by Harvey et al (2020): “ Welfare alerting indices do not directly reflect the animal’s current welfare state, but they can direct attention in future assessment towards specific animal-based indices” are related to well known and documented welfare issues, as underlined in our introduction: “In recent years, anthropogenic disturbance has had an increasing impact on the welfare of wild cetaceans around the world [1–7]. Marine debris, entanglement in fishing gear, ship collisions, underwater noise, whale-watching activities and degradation of the marine environment are currently the main welfare issues identified by international conservation bodies and marine mammal research groups [6,8–11]” (Line 40-44). Furthermore, since this is our first indicator to be validated, as mentioned in section 2: The present study is part of a larger ongoing project on the development of an integrated welfare assessment protocol for humpback and fin whales found in the Gulf of St-Lawrence, (Line 185-186) we have collected behavioural, physiological and direct environmental indices (number of whale watching boats in the area, number of cargo, number of fishing vessels, number of fishing buoys, etc. which will be presented in another article. In this regard, animal welfare alerting indices were collected but at this stage, we just wanted to present the physical state of the whales we assessed correlated with specific environmental indices (which become welfare alerting indices for future research).

Reviewer Comment References

I appreciate that I had not provided full details for all papers I suggested that the authors should look at. However, I do think the authors should have been able to find these references if a better search method had been used. Particularly, since several you have suggested have no relevance to welfare. For those that you still had not found, see below:

King et al. 2021: Assessing North Atlan?c Right Whale (Eubalaena glacialis) Welfare.

10.3390/jzbg2040052

Beaulieu 2023: Capturing wild animal welfare: a physiological perspec?ve. 10.1111/brv.13009

McMahon et al 2012: Animal welfare and decision making in wildlife research.

10.1016/j.biocon.2012.05.004

Fernandez et al 2017 (in Bu?erworth book): Pathology of Marine Mammals: What It Can Tell Us About Environment and Welfare.

Harvey et al 2021: Use of Remote Camera Traps to Evaluate Animal-Based Welfare Indicators in Individual Free-Roaming Wild Horses. 10.3390/ani11072101

Beausoleil et al. 2018: "Feelings and Fitness" Not "Feelings or Fitness"-The Raison d'etre of Conserva?on Welfare, Which Aligns Conserva?on and Animal Welfare Objec?ves.

10.3389/fvets.2018.00296

Response References

Thank you for your comment. Since we only had three days to reply to your last revision, we didn’t have time to make a full search. Thank you for providing the complete references. We have added Fernandez (Line 44) and Beausoleil (Line 78).

Finally, we would like to mention that we do recognize that our manuscript didn’t explicitly underline fundamental concepts of animal welfare science, and this was a conscious choice. The traditional approach to assess only the physical state of animals in a classical veterinary medicine /Conservation Medicine approach, is still dominating most studies with wild animals. By focusing on the physical state here in our manuscript, while pointing out how animal welfare assessment needs to be holistic (as we mentioned in our introduction), we feel we can build an applied welfare assessment step by step, with our next manuscript defining even more our protocol, including theoretical and ethical posture further by referencing other research/literature. Here, our goal was to validate, statistically, a first animal-based indicator in Mysticetes (Line 4), so we can pursue to develop our welfare assessment protocol next, by correlating other animal-based indicators (behavioural and physiological).